# Patient-specific cancer genes contribute to recurrently perturbed pathways and establish therapeutic vulnerabilities in esophageal adenocarcinoma

Thanos P. Mourikis [1,2,40], Lorena Benedetti[1,2,40], Elizabeth Foxall [1,2,40], Damjan Temelkovski[1,2], Joel Nulsen[1,2], Juliane Perner[3], Matteo Cereda[4], Jesper Lagergren[2], Michael Howell[5], Christopher Yau[6], Rebecca C. Fitzgerald[3],

Paola Scaffidi [7,8], The Oesophageal Cancer Clinical and Molecular Stratification (OCCAMS) Consortium Francesca D. Ciccarelli[1,2]

The identification of cancer-promoting genetic alterations is challenging particularly in highly unstable and heterogeneous cancers, such as esophageal adenocarcinoma (EAC). Here we describe a machine learning algorithm to identify cancer genes in individual patients considering all types of damaging alterations simultaneously. Analysing 261 EACs from the OCCAMS Consortium, we discover helper genes that, alongside well-known drivers, promote cancer. We confirm the robustness of our approach in 107 additional EACs. Unlike recurrent alterations of known drivers, these cancer helper genes are rare or patient-specific. However, they converge towards perturbations of well-known cancer processes. Recurrence of the same process perturbations, rather than individual genes, divides EACs into six clusters differing in their molecular and clinical features. Experimentally mimicking the alterations of predicted helper genes in cancer and pre-cancer cells validates their contribution to disease progression, while reverting their alterations reveals EAC acquired dependencies that can be exploited in therapy.

[1] Cancer Systems Biology Laboratory, The Francis Crick Institute, London NW1 1AT, UK. [2] School of Cancer and Pharmaceutical Sciences, King's College London, London SE11UL, UK. [3] MRC Cancer Unit, Hutchison/MRC Research Centre, University of Cambridge, Cambridge CB2 0XZ, UK. [4] Italian Institute for Genomic Medicine (IIGM), Turin 10126, Italy. [5] High Throughput Screening Laboratory, The Francis Crick Institute, 1 Midland Road, London NW1 1AT, UK. [6] University of Birmingham, Birmingham B15 2TT, UK. [7] Cancer Epigenetics Laboratory, The Francis Crick Institute, London NW1 1AT, UK. [8] UCL Cancer Institute, University College London, London WC1E 6DD, UK. [40]These authors contributed equally: Thanos P. Mourikis, Lorena Benedetti, Elizabeth Foxall. Correspondence and requests for materials should be addressed to F.D.C. (email: francesca.ciccarelli@crick.ac.uk)

Genome instability enables the onset of several hallmarks of cancer with some acquired alterations conferring selective advantages to the mutated cells and driving their outgrowth and eventual dominance[1]. The identification of driver genes (genes acquiring driver alterations) is therefore critical to fully understand the molecular determinants of cancer and to develop precision oncology. Since driver genes are under positive selection during cancer progression, a reasonable assumption is that their mutation is observed more frequently than expected. In recent years, large-scale cancer genomic studies have provided the required power to detect driver events recurring across samples with good statistical confidence[2,3]. However, the full characterisation of driver events is challenging when a cancer's genomic landscape is highly variable and recurrent events are rare.

One such cancer is esophageal adenocarcinoma (EAC), whose incidence in recent years has risen substantially in the western world[4]. EAC exhibits high mutational and chromosomal instability leading to widespread genetic heterogeneity. In over 400 EACs sequenced so far, mutations in *TP53, CDKN2A, SMARCA4, ARID1A, SMAD4, ERBB2, MYD88, PIK3CA, KAT6A, ARID2,* as well as amplifications of *VEGFA, ERBB2, EGFR, GATA4/6, CCNE1* are the most recurrent driver events[5,6]. However, a significant fraction of patients remain without known genetic determinants and the number of identified drivers per sample is too low to fully explain the disease. Consequently, the molecular mechanisms that drive EAC have been difficult to characterise in full. This impacts EAC diagnosis and treatment, with recent phase III clinical trials of targeted agents failing to show benefits or reaching inconclusive results[7,8].

Here we hypothesise that, alongside the critical role of recurrent and well-known drivers, complementary somatic alterations of other genes help cancer progression in individual patients. Therefore, the comprehensive characterisation of the full compendium of cancer drivers requires that both recurrent and rare events are considered. While recurrent drivers can be identified based on the frequency of their alterations, rare genes altered in few or even single patients are difficult to identify. To this aim, we develop sysSVM, an algorithm based on supervised machine learning that predicts cancer genes in individual patients. The rationale of sysSVM is that somatic alterations sustaining cancer affect genes with specific properties[9]. It therefore uses these properties, rather than recurrence, to identify cancer genes.

We apply sysSVM to 261 EACs from the UK OCCAMS Consortium, part of the International Cancer Genome Consortium (ICGC). We first train the classifier using 34 features derived from the biological properties specific to known cancer genes and then prioritise 952 genes that, together with the known drivers, help promote cancer development across the whole EAC cohort. The large majority of these newly predicted 'helper' genes are rare or patient-specific but they converge towards the perturbation of cancer-related processes including intracellular signalling, cell cycle regulation, proteasome activity and Toll-like receptor signalling. We use the recurrence of process perturbation, rather than genes, to stratify the 261 EACs into six clusters that show distinct molecular and clinical features and suggest differential response to targeted treatment.

## Results

**The landscape of recurrent and rare EAC genes**. sysSVM applies machine learning to predict altered genes contributing to cancer in individual patients based on the similarity of their molecular and systems-level properties to those of known cancer genes (Supplementary Note 1). Molecular properties include somatic alterations with a predicted damaging effect on the protein function (gene gains and losses, translocations, inversions, insertions, truncating and non-truncating damaging alterations and gain of function mutations) as well as the overall mutation burden and the gene copy number (Supplementary Table 1). Systems-level properties are genomic, epigenomic, evolutionary, network and gene expression features that distinguish cancer genes from other genes. They include gene length and protein domain organisation[9,10], gene duplicability[11,12], chromatin state[13], connections and position in the protein-protein interaction network[11], number of associated regulatory miRNAs[12], gene evolutionary origin[12] and breadth of gene expression in human tissues[9,10] (Supplementary Table 1).

sysSVM is composed of three steps (Fig. 1a, Supplementary Note 1). In step 1, 34 features describing the gene molecular and systems-level properties are mapped to all genes in each patient. In step 2, known cancer genes altered in the patient cohort are used to run a set of three-fold cross validations and identify the best models in four kernels (linear, sigmoid, radial, polynomial). In step 3, these best models are used for training and prediction. All altered genes except the known cancer genes used for training are first scored in each patient individually by combining the predictions of the four kernels and then ranked according to the resulting score. Since the hypothesis is that the strength of the contribution of a gene to cancer depends on how similar its properties are to those of known cancer genes, the top scoring genes in each patient are the most likely contributors to cancer progression. The overall results are combined to obtain the final list of predicted cancer genes.

We applied sysSVM to 261 EACs from OCCAMS, which are part of the ICGC dataset (Fig. 1b, Supplementary Data 1). In step 1, we extracted 17,078 genes with predicted damaging alterations (median of 382 damaged genes per patient) and mapped their 34 features. We verified that there is no pairwise correlation between these features (Supplementary Fig. 1). Moreover, 476 known cancer genes[14] altered in the 261 EACs (Supplementary Data 2) tend to cluster in distinct regions of the feature space (Supplementary Fig. 2). This confirms that these features distinguish cancer genes from other genes. In step 2, we ran 10,000 iterations of a three-fold cross validation using the 476 known cancer genes and combined the results to obtain 500 best models for each kernel (Supplementary Table 2, Methods). In step 3, we trained the four classifiers with these best models and used them to score and rank the remaining 16,602 altered genes in each patient. Since the gene score reflects a gradient between driver and passenger activity, we considered the top 10 scoring genes in each EAC as the main cancer contributors for that patient. We verified that the main findings of our study hold true if we apply higher or lower cut offs (see below). Overall, this produced 500 lists of top 10 scoring genes in each sample (Supplementary Table 2, Methods). We considered the list of 952 genes that occurred most frequently as the final set of predicted cancer genes (Supplementary Data 3). Since our hypothesis is that these genes help the known drivers to promote cancer, we define them as helper genes.

We investigated the importance of each feature in the four classifiers by ranking the 34 features based on their weight[15] and observed interesting properties of the four models. Firstly, categorical features were the top contributors for linear kernels (linear and polynomial), while both categorical and continuous features contributed to non-linear kernels (radial and sigmoid, Supplementary Fig. 3). This likely reflects intrinsic differences across kernels, and supports their integration to capture different regions of the feature space and increase the chances of identifying rare helpers. Secondly, no feature had zero weight in all four kernels, indicating that all features contributed to the final gene classification. Thirdly, despite the high prevalence of copy number gains (Supplementary Table 1), gene amplification was

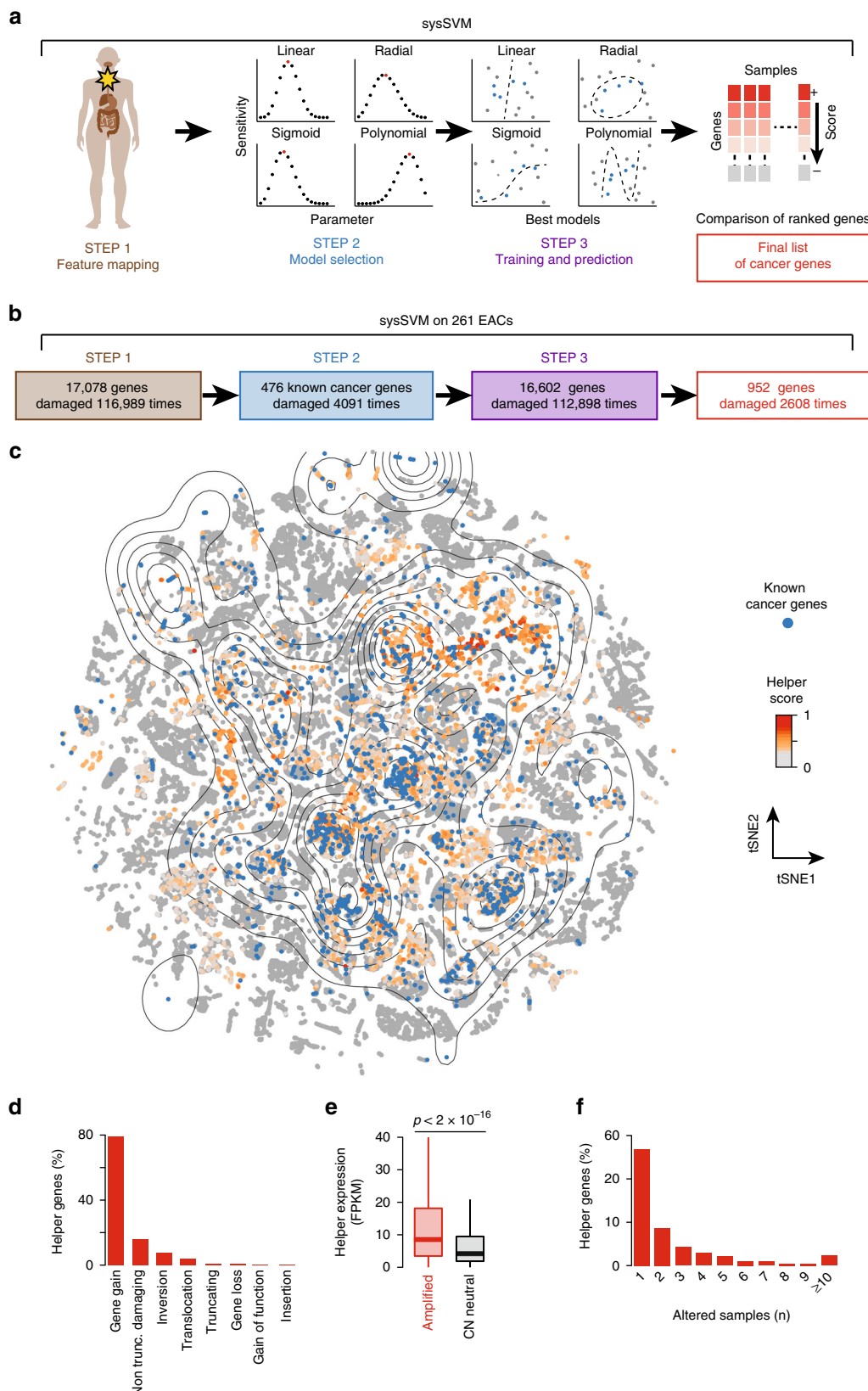

not the highest-ranking feature in any kernel (Supplementary Fig. 3). We next checked whether nodes with similar topological properties in the protein-protein interaction network had similar functional properties, since this may bias the final classification. A pathway enrichment analysis of the 2608 central hubs of the network resulted in 528 enriched pathways (FDR = 0.01), representing 46% of all Reactome pathways. This indicates a large diversity in the functions of central hubs and excludes that nodes with similar topological properties necessarily have similar functional properties. Furthermore, a pathway enrichment

**Fig. 1** Cancer helper genes in 261 EACs. **a** Schematic workflow of the sysSVM algorithm. **b** Application of sysSVM to 261 EACs. Genes with somatic damaging alterations ($n = 116,989$) were extracted from 261 EACs and divided into training (known cancer genes, blue) and prediction (rest of altered genes, purple) sets. sysSVM was trained on the properties of known drivers and the best models were used for prediction. All altered genes were scored in each patient individually and the top 10 hits were considered as the cancer helper genes in that patient, for a total of 2608 helper alterations, corresponding to 952 unique hits (red). **c** t-distributed Stochastic Neighbour Embedding (t-SNE) plot of 116,989 altered genes in 261 EACs. Starting from the 34 properties used in sysSVM, a 2-D map of the high-dimensional data was built using Rtsne package (https://github.com/jkrijthe/Rtsne) in R. Curves are coloured according to the density of 476 known cancer genes altered 4091 times (blue) used as a training set and the rest of altered genes are coloured according to their sysSVM score. **d** Distribution of damaging alterations in 952 cancer helpers. Overall, these genes acquire 2608 damaging alterations. **e** Expression of helper genes in EACs where they are amplified compared to EACs where they are copy number neutral. FPKM values from RNA-Seq were available from 92 EACs. Out of the 952 helper genes, 389 had at least one amplification across these samples, for a total of 751 amplification events. Significance was assessed using the Wilcoxon rank-sum test. Lower and upper hinges and middle line of boxplots correspond to 25th, 75th, and 50th percentiles. Upper and lower whiskers extend less than 1.5 times the interquartile range. **f** Recurrence of cancer helpers across 261 EACs. Only samples acquiring alterations with a damaging effect are considered

---

analysis only considering 252 central hubs encoded by known cancer genes resulted in 158 enriched pathways (FDR = 0.01). Of these, 135 are in common with all central hubs, indicating that cancer-related functions constitute a small fraction of pathways enriched in central hubs.

Helper genes localise to the same high-density regions of known cancer genes (Fig. 1c), with lower scoring genes being further away (Supplementary Fig. 3). The properties of top scoring genes therefore resemble those of known cancer genes. Consistent with the prevalence of gene amplification in EAC (Supplementary Table 1), the vast majority of the 952 helpers undergo copy number gain (Fig. 1d), resulting in their increased expression (Fig. 1e). Although prevalent gene amplification does not bias the best models (Supplementary Fig. 3), we investigated its impact on the final predictions. First, only a minority of all amplified genes in the 261 EACs are predicted as helper genes (Supplementary Fig. 4). Second, the 952 final helpers are amplified 6296 times in the 261 EACs, but they are only predicted as helpers 2062 times (Supplementary Fig. 4). Therefore, gene amplification alone is not sufficient for a gene to be predicted as a helper. Despite the majority of helpers being rare or patient-specific (Fig. 1f), some are altered in over 10% of EACs (Supplementary Data 3) and are usually associated with frequently occurring amplifications[16] (Supplementary Fig. 4).

We next assessed the robustness of our predictions. First, we evaluated the performance of sysSVM using two independent cohorts, 86 EACs from The Cancer Genome Atlas (TCGA) and 21 EACs from another study[17] (Supplementary Data 1). We scored all altered genes, including known cancer genes[14], in each of the 107 EACs independently, using the four best models trained on the ICGC cohort. In both datasets, known cancer genes have significantly higher scores than the rest of the altered genes (Supplementary Fig. 3), indicating that sysSVM is able to recognise them as major cancer contributors. Second, we checked whether any of the 952 helpers were previously identified as cancer genes. We found that 41 helpers (4%) have recently been added to the Cancer Gene Census[14] and 171 helpers (18%) have been predicted as candidate cancer genes in various cancers, including EAC[5] (Supplementary Data 3). Third, we searched for possible false positive predictions using two lists. The first was composed of 49 genes predicted as false positives of recurrence-based methods[5] and contained only three helpers (PCLO, CNTNAP2 and NRXN3; Supplementary Data 3). Interestingly, PCLO has recently been shown to exert an oncogenic role in esophageal cancer by interfering with EGFR signalling[18]. The second list was a manually curated set of 488 putative false positives[3] and contained 44 helpers (4.6% of the total). This is less than the fraction of known cancer genes[14] present in the same list of false positives (46/719, 6.4%). Altogether these analyses

indicate that sysSVM robustly predicts cancer genes in multiple patient cohorts, with a minimal false positive rate.

**Helper genes perturb cancer-related biological processes.** To gather a comprehensive characterisation of the molecular determinants of EAC, we analysed the biological processes perturbed by helpers compared to drivers. We manually reviewed the 476 known cancer genes[14] with damaging alterations in the OCCAMS cohort and retained 202 of them based on the concordance between the type of acquired modification and literature evidence of their role as oncogenes or tumour suppressors (Supplementary Data 2). The median number of drivers per EAC is in accordance with recent estimates[19,] with a prevalence of gene amplification (Supplementary Fig. 4). We then performed two independent gene set enrichment analyses, with either 202 known drivers or 952 helpers, to dissect their relative functional contribution to EAC. This led to 212 and 189 enriched pathways out of the 1877 tested, respectively (FDR < 0.01, Supplementary Data 4, Supplementary Fig. 4). The analysis of known drivers resulted in a higher number of enriched pathways than helpers, despite their lower number. This reflects the higher number of pathways that drivers map to (median of four pathways for drivers and two pathways for helpers).

Seventy-three pathways (over 34%) enriched in known drivers are perturbed in over 50% of EACs (Supplementary Data 4, Supplementary Fig. 5). These universal cancer pathways are involved in well-known cancer-related processes, including intracellular signalling, cell cycle control, apoptosis and DNA repair, and are associated with the most recurrently altered known drivers (TP53, CDKN2A, MYC, ERBB2, SMAD4, CDK6, KRAS; Supplementary Data 2). Interestingly, 50 of the 73 (70%) are also enriched in helpers and 86 patients with altered helpers in a universal cancer pathway have no known drivers in that pathway (Fig. 2a, Supplementary Data 4). This indicates that helpers often contribute to the perturbation of key cancer pathways and that their alteration may be sufficient for cancer development in the absence of known drivers.

Next, we clustered EACs according to the proportion of shared perturbed pathways (Methods, Fig. 2b). When using pathways enriched in known drivers, we identified five well-supported clusters (1D-5D, Fig. 2c, median silhouette score = 0.5, Supplementary Fig. 6). These clusters are clearly driven by the mutational status of the most recurrent drivers. For example, TP53 is altered in clusters 1D-3D, EGFR, ERBB2 and MYC are altered in cluster 1D and MYC and KRAS are altered in cluster 2D (Fig. 2d, Supplementary Data 1). Samples in clusters 4D and 5D show an overall lower mutational burden ($p = 0.03$, Wilcoxon rank sum test), fewer known drivers and consequently a lower number of enriched pathways ($p = 7 \times 10^{-6}$, Wilcoxon rank sum test).

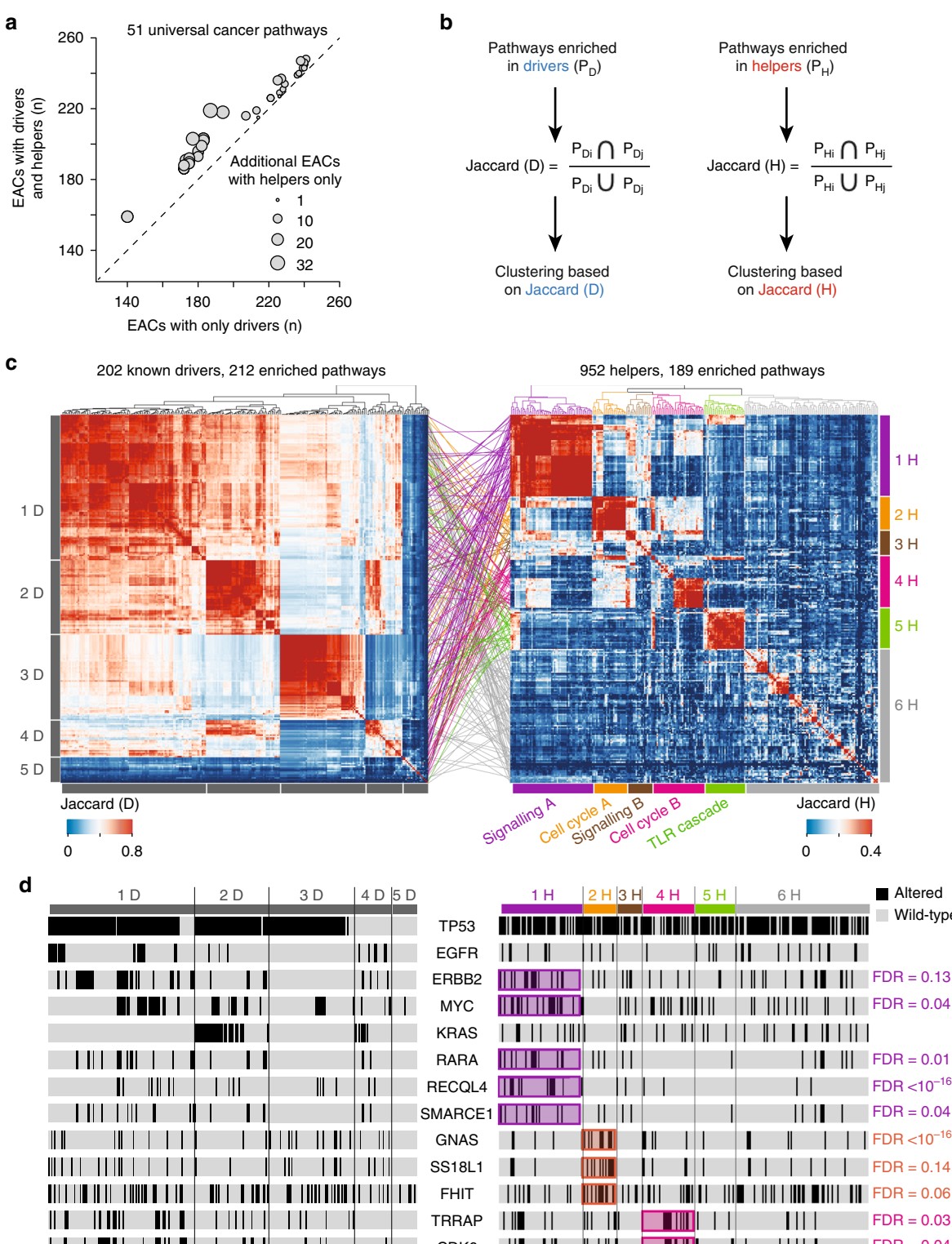

**Fig. 2** Perturbed processes in 261 EACs. **a** Scatterplot of 51 universal pathways enriched in known drivers and helpers. For each pathway, the number of EACs with altered drivers and the number of EACs with altered drivers and helpers is shown. The size of dots is proportional to the additional EACs with perturbations in these pathways because of altered helpers only. **b** Schematic of the procedure to cluster EACs according to pathways enriched in known drivers or helpers. Enriched pathways are mapped to individual EACs and the Jaccard index is calculated as the proportion of shared pathways over the total pathways in each pair of samples (*i, j*). Hierarchical clustering is then performed. **c** Clustering of 261 EACs according to pathways enriched in known drivers and helpers. Five clusters were identified using known drivers (1D-5D) and six using helpers (1H-6H). Cluster-matching coloured lines show where EACs clustered by pathways enriched in helpers map in the driver clusters. **d** Mutational status of selected known drivers across 261 EACs. Drivers enriched in clusters of helpers are highlighted. Significance was assessed using the Fisher's exact test, after correcting for False Discovery Rate (FDR)

When clustering EACs according to the pathways enriched in helpers, we identified six well-supported clusters (1H-6H, Fig. 2c, median silhouette score = 0.3, Supplementary Fig. 6). Here, samples are brought together not by the recurrent alterations of the same helpers, but by different helpers mapping to the same or related pathways (Supplementary Data 1). For example, both clusters 1H and 3H show diffuse perturbations in intracellular signalling (Fig. 2c), often involving universal cancer pathways (Supplementary Data 4, Supplementary Fig. 7). In over 43% of EACs in both clusters, perturbations in universal cancer pathways occur in patients with no drivers. Other pathways perturbed in cluster 1H, but not in 3H, involve cell cycle regulation, Toll-like receptor signalling and proteasome activity (Supplementary Data 1, Supplementary Fig. 7). EACs in cluster 1H are also significantly associated with several known drivers including *RECQL4*, *RARA*, *MYC*, *SMARCE1* and *ERBB2* (Fig. 2d), which are often but not always co-altered (Supplementary Fig. 4). They have a prevalence of mutational signature S3 and are enriched in early (stage 2) tumours (Fig. 3a). Patients in cluster 3H are instead enriched in tobacco smokers (Fig. 3a).

The processes perturbed in clusters 2H and 4H are also functionally related, in this case to cell cycle regulation (Fig. 2c). All EACs in cluster 2H have helpers involved in the regulation of the G1/S transition (Supplementary Fig. 7), including members of the E2F family of transcription factors and associated co-activators, competitors and downstream targets (Supplementary Data 1). Cluster 4H instead harbours perturbations in DNA replication, with alterations in the MCM complex, a downstream target of E2F[20]. Dysregulation of E2F transcription factors or the MCM complex can induce genomic instability through either aberrant cell-cycle control or replicative stress[21]. Indeed, EACs in clusters 2H accumulate significantly more damaged and amplified genes, while those in 4H show significantly more deletions and are enriched in mutational signature 2 (Fig. 3a). Cluster 2H also shows significant alterations of the known drivers *GNAS*, *SS18L1*, and *FHIT* (Fig. 2d). *FHIT* is linked to increased genomic instability[22] and regulates the expression of cell cycle-related genes[23], therefore potentially affecting the G1/S transition pathways of this cluster. Cluster 4H shows frequent alterations in the known drivers *TRAPP* and *CDK6* (Fig. 2d). The latter functions in various cell cycle-related pathways, including the mitotic G1/S phase pathway altered in 100% of cluster 4H (Supplementary Data 1, Supplementary Fig. 7). Interestingly, elevated expression of the MCM complex has been associated with tumour aggressiveness and poor outcome[24]. Their perturbation could therefore explain the significantly lower survival observed for patients in cluster 4H (Fig. 3b). Finally, cluster 5H shows perturbations in the Toll-like receptor (TLR) signalling cascade (Supplementary Fig. 7) that has recently been linked to EAC[25].

Overall, clusters 1H to 5H account for 166 EACs (64% of the total cohort). The remaining 95 EACs in cluster 6H share fewer perturbed pathways, although 55 of them (58%) have alterations in Rho GTPase activity (Supplementary Fig. 7) with frequent modifications of Rho GTPase effectors including *ROCK1*, *PTK2*, *PAK1*, *LIMK1* and *NDE1* (Supplementary Data 1). EACs in the six clusters obtained using helpers are broadly dispersed in the clustering of known drivers (Fig. 2c) indicating that helpers bring together patients with similar perturbed processes that cannot be appreciated when focussing only on recurrent drivers.

To test whether the germline genetic makeup of EAC patients was associated with the somatic perturbation of specific processes, we identified patients with potentially damaging germline variants in 152 known cancer predisposition genes[26]. Overall, we found 82 patients with damaging variants in 54 predisposition genes, with no significant enrichment compared to European controls[27] (Supplementary Fig. 8). This is expected since the heritable component of EAC is spread over a large number of low-impact loci[28]. We then tested for associations between cancer predisposition genes and clusters of helpers or drivers. The only significant result was a depletion of predisposition variants in cluster 4H, which is characterised by diffuse perturbations in DNA replication (Fig. 3a). Interestingly, the 54 cancer predisposition genes damaged in EAC patients are enriched in DNA repair pathways (FDR = $2.3 \times 10^{-10}$, Fisher's exact test). It is tempting to speculate that germline damages affecting DNA repair pathways would render additional somatic perturbations in the same pathways lethal and therefore be counter-selected.

Finally, we checked whether patient clustering is affected by the number of helper genes considered in each patient. We performed the same analysis considering the top five or top 15 scoring genes (528 and 1297 unique genes, respectively). We found that the vast majority (99 and 77%) of the pathways enriched in these datasets are also enriched when considering the top 10 helpers (Supplementary Fig. 8), indicating that the recurrently perturbed processes are highly overlapping. We then clustered EACs according to the proportion of shared perturbed pathways and verified that the six clusters obtained using pathways enriched in top 10 genes recapitulated well the clusters obtained using top five or 15 genes (Supplementary Fig. 8). Therefore, the clustering is robust regardless of the applied ranking cut-off.

**Helper alterations lead to cancer phenotypes and dependence.** To test the contribution of EAC helper genes to cancer, we used two experimental approaches. In the first approach, we assessed the consequences of altering (1) frequently or rarely altered helpers; (2) loss of function alterations or gain of function/ overexpression; and (3) processes that define specific helper clusters or that are altered across all helper clusters. We used diploid EAC FLO-1 cells that have no alterations in any of the helpers selected for validation[14] to allow a clear evaluation of the effect of their induced alteration. We measured cell proliferation as a main hallmark of cancer[3] and also performed gene-specific assays. In the second approach, we evaluated the dependence of EAC on helper perturbations by assessing the effect of reverting their alterations on cell growth. For this, we used EAC cell lines with alterations similar to those observed in patients.

First, we modified the most commonly altered helpers in clusters 2H and 4H, *E2F1* (23 out of 24 samples in cluster 2H) and *MCM7* (18 out of 37 samples in cluster 4H, Supplementary Data 1). Both *E2F1* and *MCM7* are amplified in EACs (Supplementary Data 3) leading to significant gene overexpression (median two-fold increase, $p = 6 \times 10^{-3}$ and $p = 8 \times 10^{-3}$, respectively, Wilcoxon rank-sum test; Fig. 4a). We therefore overexpressed *E2F1* and *MCM7* in FLO-1 cells to levels comparable to those observed in patients (Fig. 4b). In both cases, this resulted in significantly increased proliferation of over-expressing cells compared to control cells ($p = 2 \times 10^{-4}$ and $p = 9 \times 10^{-4}$, respectively, two-tailed *t*-test; Fig. 4c). Since E2F1 promotes cell cycle progression, we assessed DNA replication rate by EdU incorporation. We observed increased EdU intensity throughout S phase in *E2F1* overexpressing cells compared to control cells ($p < 10^{-4}$, Mann Whiney U test; Fig. 4d). This suggests that E2F1 may help cancer growth by promoting S phase entry. To assess the consequence of *MCM7* overexpression, we measured the loading of the MCM complex onto chromatin. *MCM7* overexpressing cells display a lower MCM fluorescence intensity overall compared to control cells when staining the chromatin-bound fraction for either MCM7 or

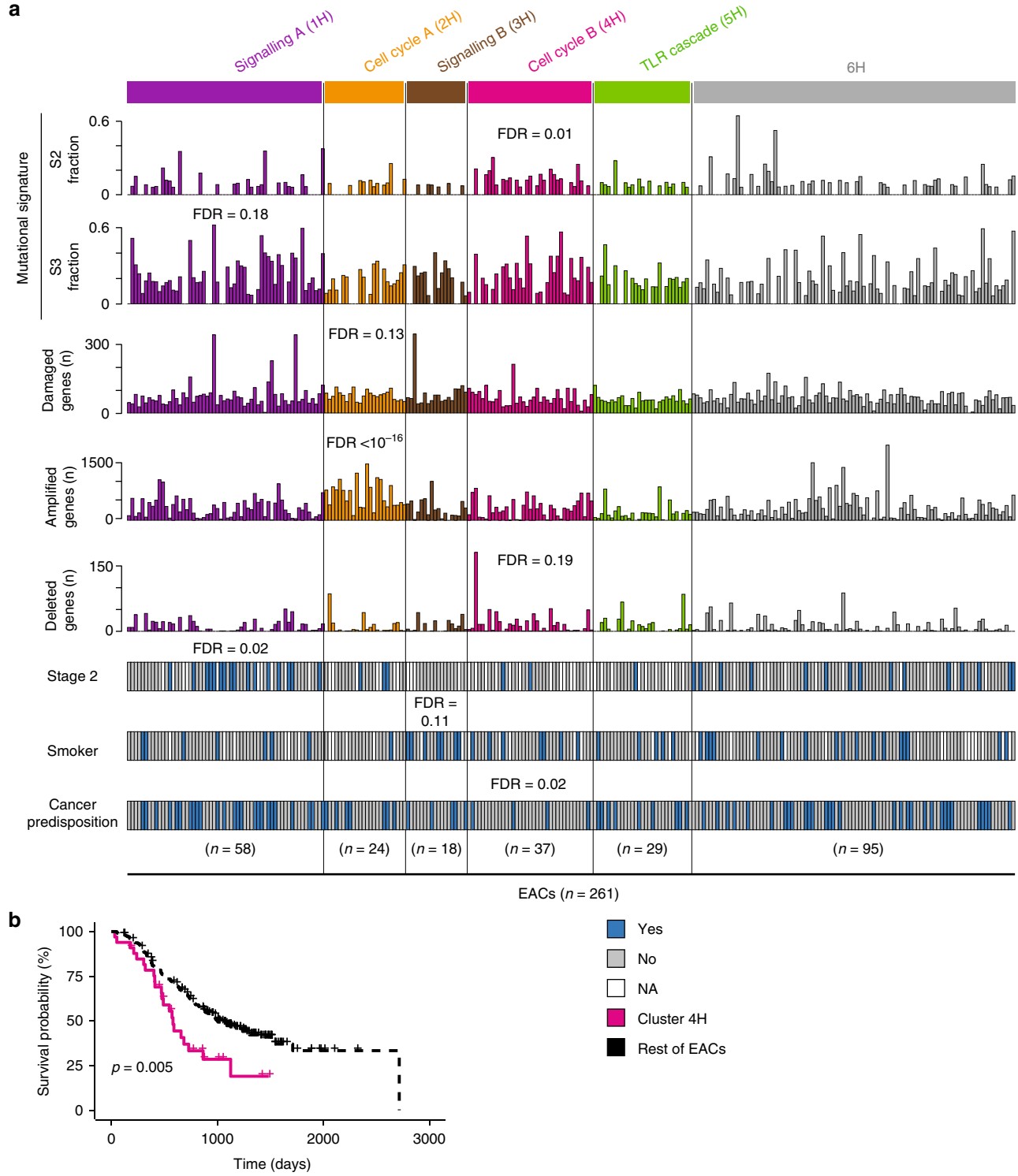

**Fig. 3** Features of EAC clusters driven by pathways enriched in helpers. **a** For each helper cluster (1H–6H) indicated are the molecular features (mutational signatures, number of genes with damaging mutations, undergoing amplification or deletion), the distribution of stage 2 tumours and the tobacco smoking habits of the patients that show significant associations with one of the six clusters of helpers. Enrichment in number of altered genes, tumour staging, smoking habits and cancer predisposition was assessed using Fisher's exact test. Distributions of mutational signatures were compared using Wilcoxon rank-sum test. FDR = false discovery rate after correction for multiple testing. **b** Kaplan–Meier survival curves of EACs in cluster 4H (n = 37) and the rest of EACs (n = 224). Analysis was performed using survival (https://github.com/therneau/survival) and survminer (https://github.com/kassambara/survminer) R packages with default parameters. Significance was measured using the log-rank test

MCM3 ($p < 10^{-4}$, Mann–Whitney U test; Fig. 4e, f). This suggests that less MCM complex is loaded onto chromatin by the start of S phase. Therefore, *MCM7* overexpression leads to both increased proliferation and perturbation of MCM complex activity. Finally,

we reduced *MCM7* expression levels in MFD-1 cells derived from one of our EAC patients[29]. MFD-1 cells have four-fold higher *MCM7* basal expression compared to FLO-1 cells (Fig. 4g). We therefore used a doxycycline-inducible shRNA lentiviral vector

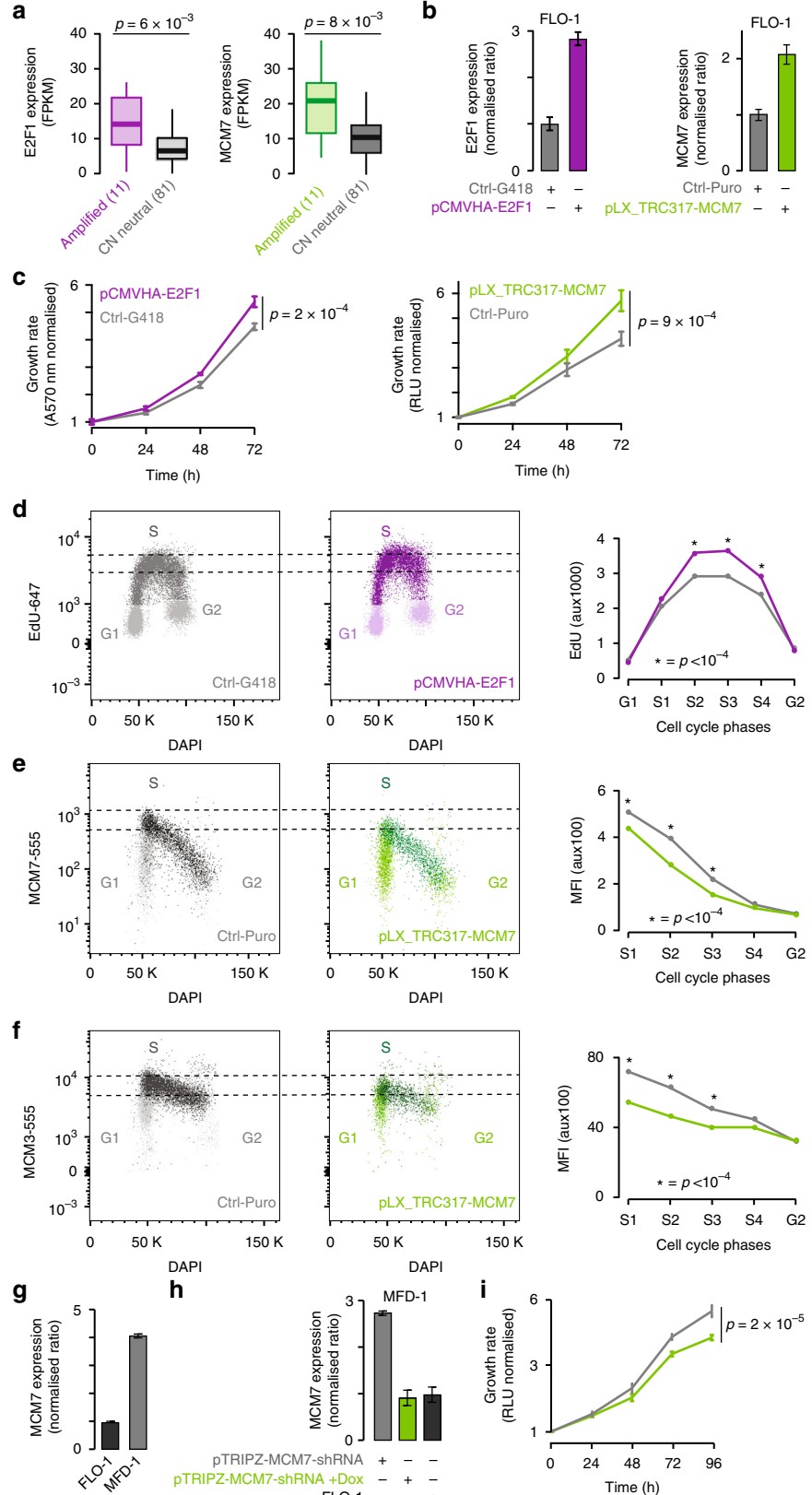

(Supplementary Table 3) to reduce *MCM7* expression in MFD-1 cells to the level of FLO-1 cells (Fig. 4h). This led to a significant decrease in proliferation ($p = 2 \times 10^{-5}$, two-tailed *t*-test; Fig. 4i), indicating that MFD-1 cells rely on *MCM7* overexpression for their growth.

Next, we evaluated the role of rarely altered helpers, such as *NCOR2* that is altered in eight EACs across five of the six clusters (Supplementary Data 3). NCOR2 contributes to the nuclear receptor corepressor complex that favours global chromatin deacetylation and transcriptional repression[5,30] (Fig. 5a).

**Fig. 4** Cancer helper role of *E2F1* and *MCM7*. **a** *E2F1* and *MCM7* expression in EACs with amplification ($n = 11$ samples each) compared to copy number neutral EACs ($n = 81$ samples each). Significance was assessed by Wilcoxon rank-sum test. **b** *E2F1* and *MCM7* mRNA expression in FLO-1 cells. Lower and upper hinges and middle line of boxplots correspond to 25th, 75th, and 50th percentiles. Upper and lower whiskers extend less than 1.5 times the interquartile range. **c** Proliferation of FLO-1 cells overexpressing *E2F1* or *MCM7* compared to control. **d**. EdU (5-ethynyl-2′-deoxyuridine) incorporation by flow cytometry in *E2F1* overexpressing cells compared to control. Cells were separated into G1, S and G2 phases. S phase cells were subdivided into 4 gates (S1-S4, Supplementary Fig. 9). Differences in EdU geometric mean fluorescence intensity were assessed by Mann-Whitney U test. MCM complex loading onto chromatin in *MCM7* overexpressing or control cells through MCM7 **e** or MCM3 **f** staining. Cells were pulsed with EdU and chromatin was fractioned before staining for MCM7 or MCM3 to detect chromatin-bound MCM complex. Cells were separated into cell cycle phases using EdU and DAPI intensity. MCM7 or MCM3 fluorescence intensity during S phase illustrates MCM complex unloading from chromatin. Differences in geometric mean fluorescence intensity of MCM staining were assessed using Mann-Whitney U test. For (d), (e), (f) one representative of $n = 3$ biological replicates is shown. Corresponding pseudocolour plots are in Supplementary Fig. 9. **g** *MCM7* mRNA expression in MFD-1 and FLO-1 cells. **h** qRT-PCR *MCM7* expression in MFD-1 cells after transduction with a *MCM7*-shRNA inducible lentiviral vector. Expression was assessed with or without 96 h doxycycline treatment. **i** Proliferation curve of MFD-1 cells with or without doxycycline-induced *MCM7* knockdown. For all qRT-PCR experiments, expression was relativised to β-2-microglobulin and normalised to FLO-1 cells. $N = 2$ biological replicates were performed, each in technical triplicate. For all proliferation assays, $n = 2$ biological replicates were performed, each with four technical replicates. Proliferation was assessed every 24 h and normalised to time zero. Mean values at 72 h were compared by two-tailed Student's *t*-test. Error bars show standard deviation. Source data are available in the Source Data file

Consistently with the suggested tumour suppressor role of *NCOR2* in lymphoma and prostate cancer[31], the most frequent *NCOR2* alterations in EAC lead to a loss of function. To reproduce these alterations, we edited *NCOR2* in FLO-1 cells using a vector-free CRISPR system[32] (Methods, Supplementary Table 3) and the editing was quantified using MiSeq (Fig. 5b). We observed a 1.3-fold increase in proliferation of edited cells compared to the control cells ($p = 3 \times 10^{-3}$, two-tailed *t*-test test; Fig. 5c).

Then, we tested the effect of altering members of the Rho GTPase effector pathway, pervasively perturbed in all six clusters, often through patient-specific alterations (Fig. 5d, Supplementary Data 1). We modified *ABI2* and *PAK1*, which undergo damaging alterations and amplification in one and nine EACs, respectively (Supplementary Data 3). We therefore edited *ABI2* and over-expressed *PAK1* as described above (Supplementary Table 3, Fig. 5e) and observed significantly increased proliferation compared to control cells (*ABI2*: $p = 4 \times 10^{-4}$, *PAK1*: $p = 1 \times 10^{-3}$ two-tailed *t*-test; Fig. 5f).

Finally, we focussed on *PSMD3* encoding a subunit of the regulatory 19S proteasome complex. *PSMD3* is amplified and overexpressed in three EACs of cluster 1H, which overall contains 14 samples with alterations in six proteasome subunits (Fig. 6a and Supplementary Data 3). Three EAC cell lines (MFD-1, OE19 and OE33) show higher basal expression of *PSMD3* compared to FLO-1 (2-, 3- and 4-fold increase respectively, Fig. 6b). Using a doxycycline-inducible lentiviral shRNA vector (Supplementary Table 3), we reduced *PSMD3* expression in MFD-1, OE19 and OE33 cells to levels equivalent to those of FLO-1 (Fig. 6c). In all three cell lines we observed a significant reduction in cell proliferation following the reduction of *PSMD3* expression (MFD-1: $p = 4 \times 10^{-8}$; OE19: $p = 2 \times 10^{-8}$; OE33: $p = 6 \times 10^{-3}$, two-tailed *t*-test; Fig. 6d). The effect was particularly strong in OE19, where cell growth was arrested completely. MFD-1 and OE33 showed 1.3- and 1.2-fold reductions in cell growth (Fig. 6d). This suggests that the extent of EAC reliance upon helper alterations is at least partially context dependent.

Taken together, our experimental data indicate that, independently of the alteration frequency, the modification of helpers positively affects EAC cell growth. The fold changes in proliferation rate observed upon perturbation of helpers are in the same range as those observed following alteration of known strong drivers including TP53 or PIK3CA[33,34]. Moreover, we provide evidence that EAC cells become addicted to helper alterations, suggesting that targeting helpers, or the pathways in which they act, could reduce EAC progression.

**Helper alterations promote growth in Barrett's esophagus**. To evaluate the role of helper perturbations in the early stages of EAC, we quantified their alterations in 82 samples of Barrett's esophagus (BE)[35], a pre-malignant condition associated with EAC. When considering both damaging and non-damaging alterations, the percentages of helpers and known drivers[14] altered in the whole BE cohort were comparable to that of cancer-unrelated genes (Fig. 7a). However, when considering only damaging alterations, helpers showed significant enrichment compared to the rest of genes ($p = 4 \times 10^{-13}$ and $p = 3 \times 10^{-6}$, respectively, Fisher's exact test; Fig. 7b). Early alteration of helpers in pre-malignant lesions suggests that they may favour the transition to cancer. To further validate this hypothesis, we altered representative helpers in BE CP-A cells and evaluated the impact of their alterations on cell growth. As already reported[6,35], damaging point mutations were the most common alterations in BE (Fig. 7c). Therefore, we edited *ABI2* and *NCOR2* to mimic their putative loss-of-function alterations. Since CP-A cells have wild-type *TP53*[36], we also edited *TP53* to compare the effect of altering helpers to that of altering a strong driver. After confirming high editing levels (Fig. 7d), we observed a significant increase in cell proliferation compared to control cells in all three cases (*ABI2*: $p = 3 \times 10^{-4}$, *NCOR2*: $p = 1 \times 10^{-2}$, *TP53*: $p = 1 \times 10^{-4}$, two-tailed *t*-test; Fig. 7e). Strikingly, both helpers promote cell growth to comparable levels to *TP53*, suggesting that helper alterations can favour early cancer progression to a similar extent to driver alterations.

## Discussion

Most state-of-the-art approaches to discovering cancer driver events rely on the detection of positively selected alterations of genes that promote cancer development[3,19]. Even ratiometric methods based on gene properties[37] ultimately assess the effect of positive selection and distinguish the few selected drivers from the many passenger events. As a result, the discovery of cancer drivers is biased towards frequently altered genes, with significant limitations for cancers such as EAC that have a highly variable but mostly flat (*i.e.* with few recurrent events) mutational landscape. Indeed, the overall selection acting on esophageal cancer genomes is among the lowest across cancer types[19], despite a median of 382 damaged genes per EAC (Supplementary Data 1). Therefore, the exclusive focus on genes under strong selection is likely to return only a partial representation of the genes involved in EAC.

To overcome these limitations, our machine learning approach sysSVM ranks somatically altered genes relevant to cancer development based on their properties rather than mutation recurrence. sysSVM also considers all types of gene alterations

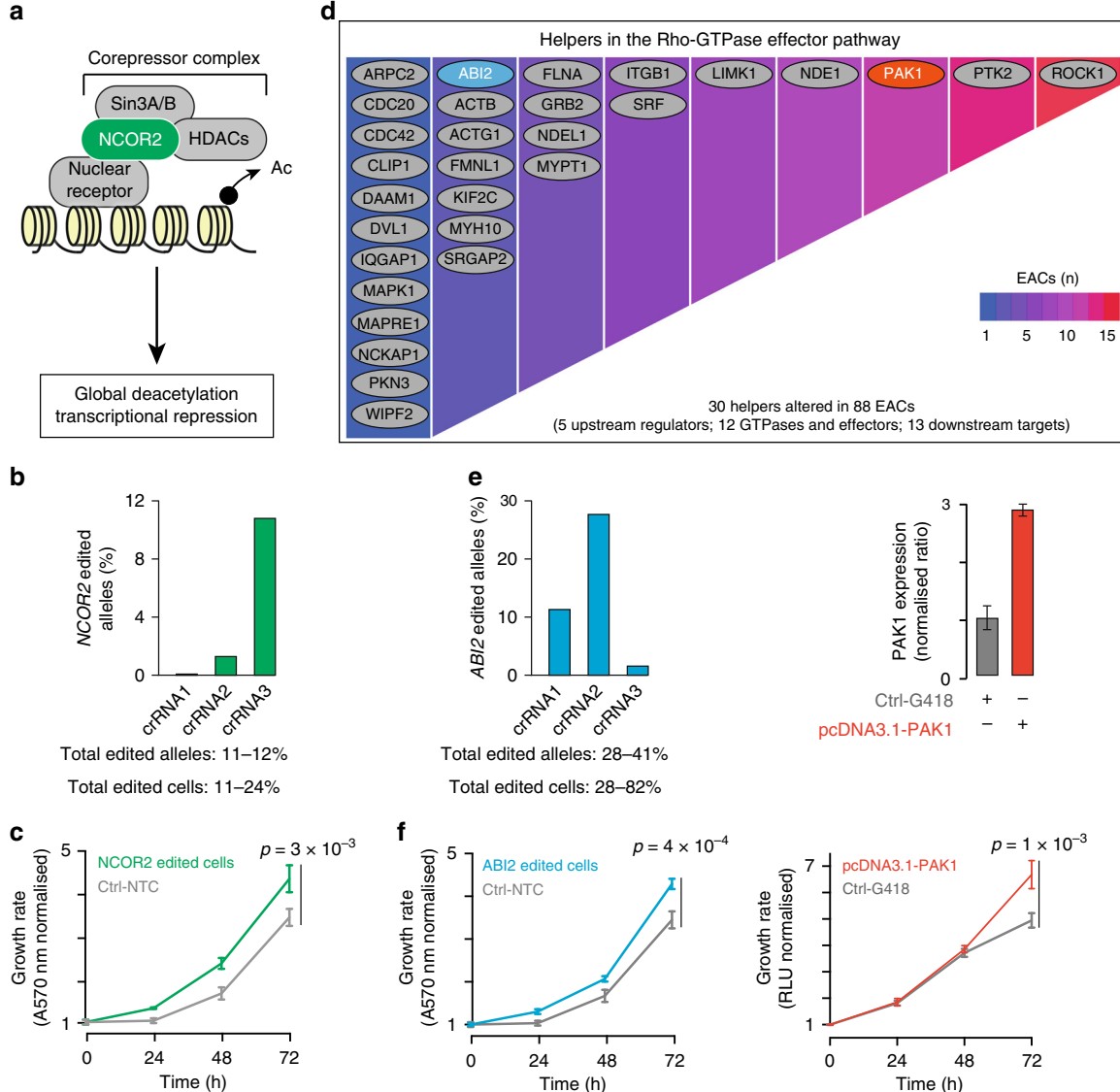

**Fig. 5** Cancer helper role of selected genes. **a** Function of NCOR2 as part of the nuclear receptor co-repressor complex, whose activity results in chromatin deacetylation and transcriptional repression. **b** Editing of the *NCOR2* gene using three pooled crRNAs where cells are transiently co-transfected with Cas9 protein, crRNAs and tracrRNA[32]. The editing efficiency was measured using MiSeq. For *NCOR2* two regions were sequenced, one containing the region targeted by crRNA1 and 2 and the other the region targeted bycrRNA3. For *ABI2* only one amplicon was sequenced, containing the region targeted by all three crRNAs. The range of edited alleles and edited cells was derived considering the two opposite scenarios where all three crRNAs edit the same alleles/cells or different alleles/cells, respectively. **c** Proliferation curve of *NCOR2* or non-targeting control (NTC) edited FLO-1 cells. **d** Manual curation of the helpers contributing to the Rho-GTPase effectors pathway. Heatmap indicates the number of samples with alterations in each gene. ABI2 (blue) and PAK1 (red) were selected for experimental validation. **e** Induced alterations in *ABI2* and *PAK1* genes. Editing of *ABI2* was performed and assessed as described for *NCOR2*. *PAK1* mRNA expression in FLO-1 cells was assessed by qRT-PCR, relativised to β-2-microglobulin and normalised to control cells. Experiments were done in triplicate in *n* = 2 biological replicates. **f** Proliferation curves of FLO-1 cells after *ABI2* editing or *PAK1* overexpression. For all proliferation assays, *n* = 3 biological replicates were performed, each with four technical replicates. Proliferation was assessed every 24 h and each time point was normalised to time zero. Mean values at 72 h were compared by two-tailed Student's *t*-test. All error bars show standard deviation. Source data are available in the Source Data file

(SNVs, indels, CNVs, and structural variations) simultaneously, providing a comprehensive overview of the genetic modifications with a cancer role in individual patients. When applied to 261 EACs, sysSVM prioritises 952 altered genes that, together with known drivers, help cancer progression. The large number reflects the positive correlation between mutational burden and number of driver genes, which is only partially explained by a sample size effect[3]. This positive correlation may indicate that the number of functionally relevant genes increases with the number of altered genes.

The heterogeneous landscape of EAC is substantially reduced by considering the perturbed biological processes (Fig. 2c). Most of these processes are well-known contributors to cancer development, including intracellular signalling, cell cycle control, and DNA repair (Supplementary Data 4). Interestingly, while the known drivers tend to encode upstream players in these pathways, helpers are often downstream effectors. For example, we found several Rho GTPase effectors (Fig. 5d, Supplementary Data 3) or genes downstream of previously reported EAC drivers in the Toll-like receptor cascade (Supplementary Data 3). This

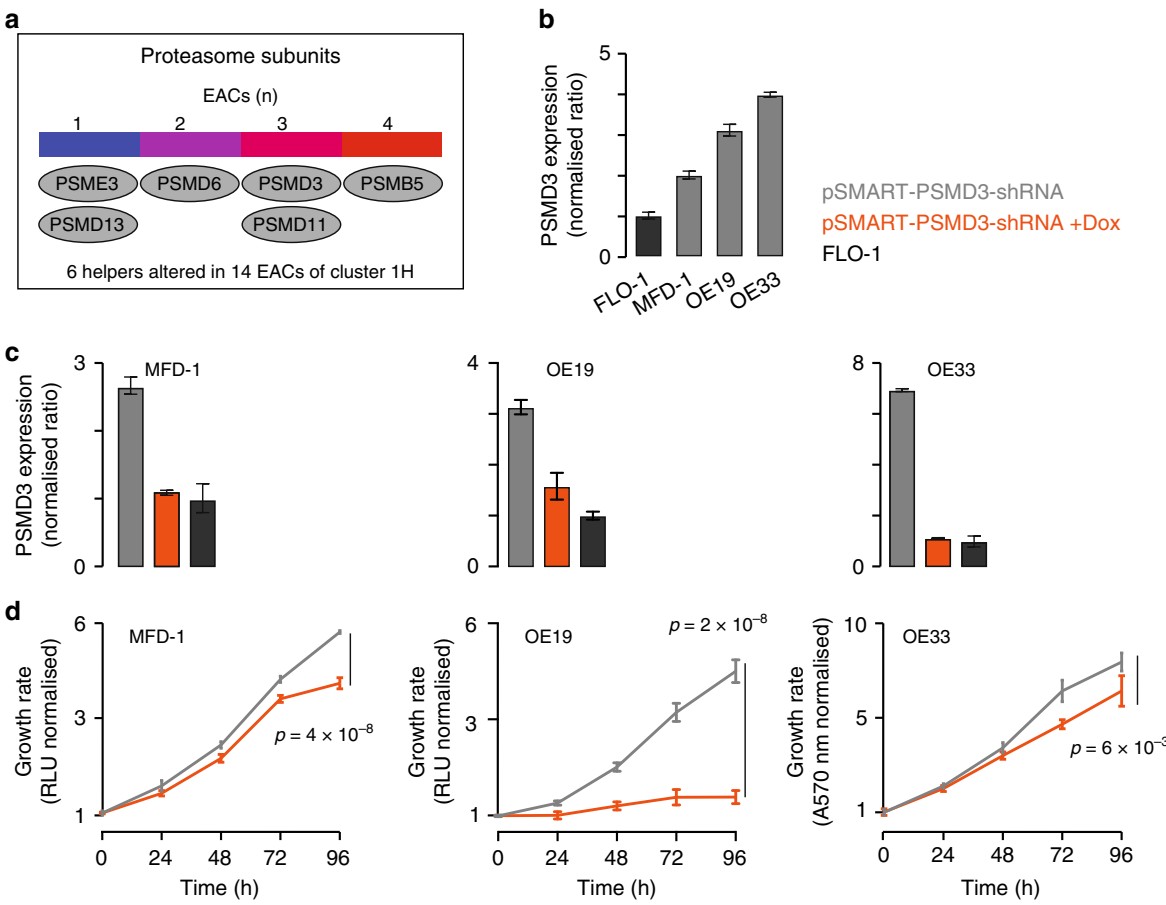

**Fig. 6** EAC cell dependence on *PSMD3* alteration. **a** Heatmap of proteasome subunits predicted as helpers in 261 EACs. **b** *PSMD3* basal mRNA expression levels in FLO-1, MFD-1, OE19 and OE33 cells. Expression was relativised to β-2-microglobulin and normalised to FLO-1 cells. **c** *PSMD3* expression levels in MFD-1, OE19 and OE33 after transduction with a lentiviral vector carrying an inducible shRNA against *PSMD3*. Expression was assessed in the absence of doxycycline and after 96 h of doxycycline treatment, relativised to β-2-microglobulin and normalised to FLO-1 cells. **d** Proliferation curves of MFD-1, OE19 and OE33 cells with or without doxycycline treatment to reduce *PSMD3* expression to levels comparable to those of FLO-1 cells. For all proliferation assays, $n = 2$ biological replicates were performed, each with four technical replicates. Proliferation was assessed every 24 h and each time point was normalised to time zero. Mean values at 96 h were compared by two-tailed Student's *t*-test. All error bars show standard deviation. Source data are available in the Source Data file

supports a more local role of helpers in contributing to cancer at the single patient level, through sustaining or complementing driver functions. In this respect, helpers are conceptually similar to mini-drivers[38].

Clustering pathways disrupted by helpers allows the division of the 261 EACs into six clusters that are often functionally related. For example, clusters 1H and 3H share perturbations in intracellular signalling. Similarly, clusters 2H and 4H show perturbations in cell cycle regulation, including S phase entry and DNA replication. Consistent with this, they bring together the most genomically unstable samples. By experimentally mimicking the amplification of *E2F1* (representative of cluster 2H) and *MCM7* (representative of cluster 4H), we increased proliferation in EAC cells (Fig. 4c). We also provide evidence that E2F1 increases proliferation by promoting S phase entry (Fig. 4d). Interestingly, *MCM7* overexpression reduced MCM complex loading onto chromatin (Fig. 4e, f), maybe due to a stoichiometric imbalance of complex subunits. This may indicate that MCM7 promotes cell growth through a separate mechanism besides its function in the MCM complex. For example, MCM7 interacts with the tumour suppressor protein Rb, a well-characterised inhibitor of E2F1[39]. It is possible that *MCM7* overexpression may sequester Rb away from E2F1, thereby promoting E2F1-mediated cell cycle progression.

Moreover, reducing *MCM7* expression in cells with high basal expression led to decreased cell proliferation, showing the dependence of cancer cells on this alteration.

We also confirmed the cancer promoting role of very rare helpers, including *ABI2*, *NCOR2* and *PAK1* that are altered in 1–4% of EACs (Fig. 5c–f). Therefore, irrespective of the frequency across patients, helpers have a substantial impact on cancer progression. We therefore speculate that it is the contribution of several genes perturbing the same pathways that promotes cancer progression rather than the alteration of one gene alone. In line with this, the dysregulation of a strong driver such as *TP53*[33] or *PIK3CA*[34] alone does not have a dramatic effect. This is confirmed by our data where the alteration of one helper gene has a mild yet significant effect on cell proliferation. Interestingly, helpers are also frequently altered in pre-cancerous lesions known as Barrett's esophagus (Fig. 7b), indicating that their alteration may be an early event in EAC transformation. Consistent with this, the perturbation of helpers leads to increased proliferation in BE cells and the effect is comparable to that of perturbing *TP53* (Fig. 7e).

The expansion of the repertoire of cancer genes may indicate new, patient-specific gene dependencies suggesting possible stratifications that could inform the selection of targeted treatments. For example, 14 samples of cluster 1H have alterations of

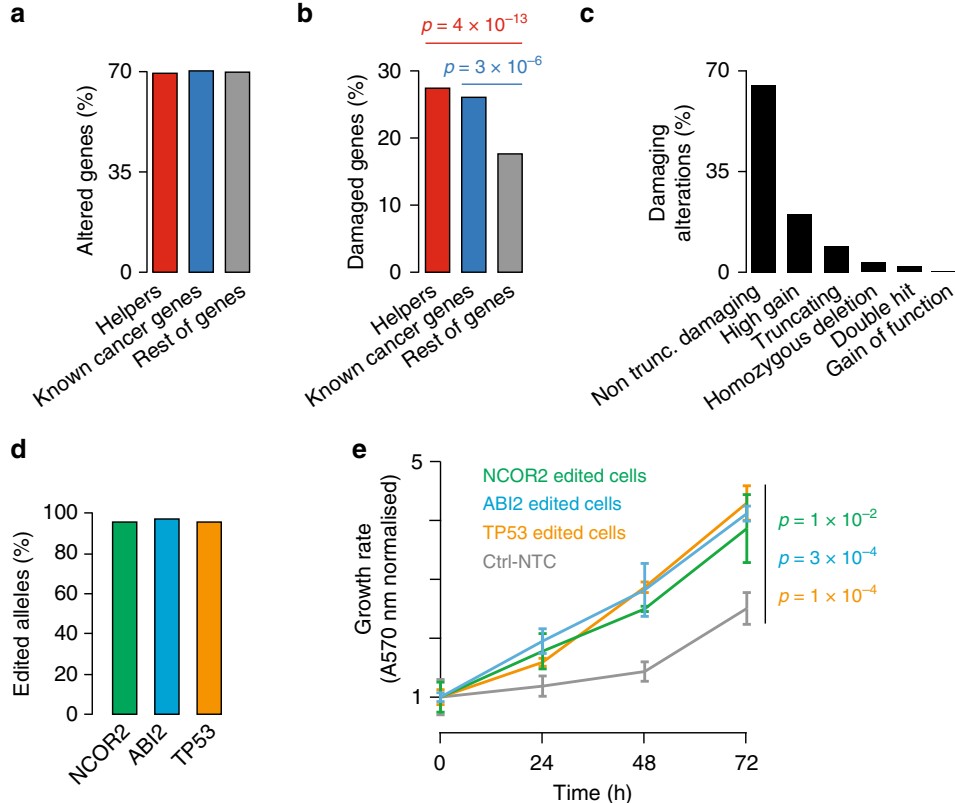

**Fig. 7** Effect of helper alterations in Barrett's esophagus cells. **a** Percentages of altered helper genes (661/952), known cancer genes (364/518) and rest of cancer unrelated genes (12,250/17,544) over the corresponding total. **b** Percentages of damaged helper genes (261/952), known cancer genes (135/518) and rest of cancer unrelated genes (3,093/17,544) over the corresponding total. Statistical significance was assessed with two-tailed Fisher's exact test. **c** Distribution of 4862 damaging alterations in 3489 damaged genes in 82 samples of BE. The damaging effect of BE variants was assessed as in EAC (Methods) with the exception of gene gains. In this case, only 'high gains' defined as in the original publication[35] were considered as damaging. **d** Editing of *ABI2*, *NCOR2* or *TP53* in CP-A cells through nucleofection of two crRNAs and Cas9 protein. The editing efficiency was measured by MiSeq and quantified as described above. For all three genes, a single amplicon was sequenced. *NCOR2*, *ABI2* and *TP53* crRNAs overlapped for over 70% of their sequences and therefore it was not possible to assess their editing efficiency independently. **e** Proliferation curves of CP-A cells after *ABI2*, *NCOR2* or *TP53* editing compared to control cells. For all proliferation experiments, n = 3 biological replicates were performed, each with four technical replicates. Proliferation was assessed every 24 h and each time point was normalised to time zero. Mean values at 72 h were compared by two-tailed Student's t-test. Error bars show standard deviation. Source data are available in the Source Data file

several proteasome subunits (Fig. 6a, Supplementary Data 1). Experimentally reverting the expression of the proteasome subunit PSMD3 to diploid levels reduced proliferation in three different EAC cell lines (Fig. 6d), suggesting that EACs become addicted to helper alterations and vulnerable to their inhibition. Interestingly, proteasome inhibition has been shown to have a synergic effect in combination with ERBB2 inhibitors[40]. Since *ERBB2* is also significantly altered in cluster 1H (Fig. 2d), a combined therapy may be beneficial to patients in this cluster.

In summary, we provide an attempt to extend the discovery of acquired perturbations contributing to cancer beyond those of recurrent drivers. Additional efforts are required to fully exploit the potential of these approaches to offer a more comprehensive view of the molecular mechanisms behind cancer and to guide clinical interventions.

## Methods

**Annotation of molecular properties**. Data on somatic single nucleotide variations (SNVs), small insertions and deletions (indels), copy number variations (CNVs), structural variations (SVs), and mutational signatures for 261 EACs were obtained from ICGC and analysed as previously described[16] (Supplementary Data 1). Briefly, SNVs and indels were called using Strelka v.1.0.13[41] and subsequently filtered[16]. For CNVs, the absolute copy number for each genomic region was obtained from ASCAT-NGS v.2.1[42] after correction for tumour content, using read

counts at germline heterozygous positions as derived from GATK v.3.2–2[43]. To account for the high number of amplifications occurring in EAC, copy number gains were corrected by the ploidy of each sample as estimated by ASCAT-NGS. A gene was assigned with the copy number of a CNV region if at least 25% of its length was contained in that region. SVs (gene translocations, inversions, insertions) were identified from discordant read pairs using Manta[44] after excluding SVs that were also present in more than two normal samples of a panel of 15 esophagus and 50 blood samples[16]. In the case of the TCGA validation cohort, SNVs, indels, and CNVs were derived from level 3 TCGA annotation data of 86 EACs (Supplementary Data 1). In the case of 21 EACs from a previous study[17], SNVs, indels, and CNVs were called as described for the ICGC samples (Supplementary Data 1). The distribution of variant allele frequency of SNVs and indels across all samples was used to remove outliers likely indicating sequencing or calling artefacts. Variants with < 10% frequency and indels longer than five base pairs were also removed. For CNVs, genomic regions were considered as amplified or deleted if their segment mean was higher than 0.3 or lower than −0.3, respectively, capping the segment mean to 1.5 to avoid hypersegmentation[45]. A gene was considered as amplified or deleted if at least 25% of its length was contained in a CNV region and the resulting copy number (CN) was estimated as:

$$CN = 2 \times 2^{segment\ mean} \qquad (1)$$

No SV data were available for the validation cohorts.

Since only genes with predicted damaging alterations were used as input for sysSVM, further annotation for the variant damaging effect was performed. Stopgain, stoploss, frameshift, nonframeshift, nonsynonymous, and splicing SNVs and indels were annotated using ANNOVAR (December 2015)[46]. All truncating alterations (stopgain, stoploss, and frameshift mutations) were considered as damaging. Nonframeshift and nonsynonymous mutations were considered as

non-truncating damaging alterations if predicted by at least five of seven function-based methods (SIFT, PolyPhen-2 HDIV, PolyPhen-2 HVAR, MutationTaster, MutationAssessor, LRTand FATHMM) or by two out of three conservation-based methods (PhyloP, GERP + + RS, SiPhy), using the scores from dbNSFP v.3.0[47]. Splicing modifications were considered as damaging if predicted by at least one of the two ensemble algorithms as implemented in dbNSFP v3.0. Putative gain of function alterations were predicted with OncodriveClust[48] with default parameters and applying a false discovery rate of 10%. The transcript lengths to estimate mutation clustering were derived from the refGene table of UCSC Table Browser (https://genome.ucsc.edu/cgi-bin/hgTables). Gene gains, homozygous losses, translocations, inversions, insertions were always considered as putative damaging alterations.

Overall, 17,078 genes had at least one damaging alteration, for a total of 116,989 redundant damaged genes across 261 EACs (Supplementary Table 1). Of these, 476 were known cancer genes[14], corresponding to 4091 redundant genes (Supplementary Data 1). For all 17,078 genes, the total number of exonic alterations (silent and nonsilent) and the somatic copy number were used as additional molecular features in sysSVM.

**Annotation of systems-level properties.** Protein sequences from RefSeq v.63[49] were aligned to the human reference genome assembly GRCh37 to define unique gene loci[10]. The length of the longest coding sequence was taken as the gene length. Genes aligning to more than one gene locus for at least 60% of the protein length were considered as duplicated genes[11]. Data on human ohnologs (gene duplicates retained after whole genome duplications) were collected from Makino et al.[50]. The number of protein domains was derived from CDD[51]. The gene chromatin state based on Hi-C experiments[13] was retrieved from the covariate matrix of MutSigCV v1.2.01[2]. Data on protein-protein and miRNA-gene interactions, gene evolutionary origin and gene expression were retrieved as described in An et al.[10]. Briefly, the human protein-protein interaction network was rebuilt from the integration of BioGRID v.3.4.125;[52] MIntAct v.190;[53] DIP (April 2015);[54] HPRD v.9;[55] the miRNA-gene interactions were derived from miRTarBase v.4.5[56] and miRecords (April 2013);[57] gene evolutionary origin was assessed as described in D'Antonio et al.[12] using gene orthology from EggNOG v.4;[58] and gene expression in 30 normal tissues was retrieved from GTEx v.1.1.8[59]. Except gene length, duplication and ohnologs, all other systems-level properties had missing information for some of the 17,078 altered genes (Supplementary Table 1). To account for this, median imputation for continuous properties and mode imputation for categorical properties were implemented. Specifically, for each property median or mode values were calculated for known cancer genes and the rest of mutated genes. All missing values were replaced with their corresponding median or mode values.

**Application of sysSVM to EACs.** The three steps of sysSVM were applied to 261 EACs (Fig. 1a, b, Supplementary Note 1). In step 1, all 34 features derived from molecular and systems-level properties (Supplementary Table 1) were mapped to the 17,078 altered genes in the cohort. Each feature was scaled to zero mean and unit variance to correct for the different numerical ranges across them. In step 2, 476 known cancer genes with damaging alterations (Supplementary Data 2) were used as a set of true positives for model selection. To optimise the parameters of the four kernels (linear, radial, sigmoid and polynomial) a grid search using 10,000 iterations of a three-fold cross validation was performed. At each iteration, the 476 known cancer genes were randomly split into 2/3 (around 317 genes) used as a training set and 1/3 (around 159 genes) used as the test set. At each increment of 100 cross validation iterations, the four best models (one per kernel) were chosen based on the median and variance of the sensitivity distribution across all previous iterations of cross-validation. The selection of the 100 sets of best models from all 10,000 cross-validation iterations was repeated 5 times, where all iterations were randomly re-ordered. In step 3, the resulting 500 best models were trained with the whole training set and used to rank the remaining 16,602 unique genes in each patient. A score was measured to combine the predictions from the four kernels and the genes not expressed in normal esophagus according GTEx annotation were excluded. These produced 500 lists of top 10 genes. Out of 500 best models, 38 had a unique set of parameters resulting in 24 unique lists of top 10 genes (Supplementary Table 2). These 24 lists ranged between 898 and 952 genes, with a core set of 598 genes shared across all of them. The most frequent top 10 list occurred 207 times (952_A, 41.4%, Supplementary Table 2). It was followed by 952_B (32.2%, 161 times) and 951_A (8.6%, 43 times). These three lists accounted for 82.2% of the 500 sets of top 10 genes, they shared 950 genes and were predicted by models differing in only one parameter (gamma in the polynomial kernel, Supplementary Table 2). Furthermore, the most frequent list was always predicted by the same set of best models. Therefore, 952_A represented a robust set of prediction and was considered as the final list of helper genes (Supplementary Data 3). To quantify the relative contribution of the 34 features to the four best models used to predict the final set of helper genes in the 952_A final list, Recursive Feature Elimination (RFE)[15] was implemented. RFE first defines the best set of parameters for each kernel. Secondly, RFE trains the one-class classifier in each kernel and computes the weight ($w$) of each feature, defined as the product of the sysSVM coefficients with each support vector. Thirdly, RFE ranks the features according to $w^2$ and recursively removes the feature with the smallest value of $w^2$ until no feature

remains. In the case of sysSVM this was done 34 times until all sysSVM features were ranked (Supplementary Fig. 3).

**Identification of perturbed processes and patient clustering.** To identify the perturbed biological processes in the EAC cohort, both predicted cancer helper genes and known cancer genes were used. A manual revision of 476 known cancer genes altered in the ICGC cohort was performed and genes were considered as known drivers if (a) their somatic alteration had been previously associated with EAC, (b) they had a loss-of-function alteration and their tumour suppressor role had been reported in other cancer types[60], (c) they had a gain-of-function alteration and their oncogenic role had been reported in other cancer types[60]. The resulting 202 known cancer drivers (Supplementary Data 2) and 952 cancer helpers were used for the gene set enrichment analysis against Reactome v.58[61], composed of 1877 pathways and 10,131 genes. After excluding pathways in levels 1 and 2 of Reactome hierarchy and those with less than 10 or over 500 genes, 1155 pathways were retained. These contained 9061 genes, including 155 known drivers and 648 helpers. Gene set enrichment was assessed using a one-sided hypergeometric test and the resulting $P$ values were corrected for multiple testing using the Benjamini & Hochberg method (Supplementary Data 4). Enriched pathways within the sets of known drivers or helpers were subsequently used to cluster samples taking into account the proportion of perturbed processes shared between samples. The Jaccard index ($A$) was calculated by deriving the proportion of shared perturbed processes between all possible sample pairs as:

$$A_{ij} = \left| P_i \cap P_j \right| / \left| P_i \cup P_j \right| \quad (2)$$

where $P_i$ and $P_j$ are the perturbed processes in samples $i$ and $j$, respectively.

Complete linkage hierarchical clustering using Euclidean distance between each row was performed on the resulting matrix. Clusters were visualised using ComplexHeatmap R package[62]. To identify the optimal number of clusters, the median silhouette value of the samples for between 3 and 20 clusters was measured as a measure of clustering robustness[63].

**Annotation of germline variants.** Starting from germline variants (SNVs and indels)[16], a series of filters were applied. First, heterozygous calls with Variant Allele Frequency (VAF) deviating from 50% by more than 2.6 standard deviations and homozygous calls with VAF less than 95% were removed. Second, variants were removed if their Minor Allele Frequency (MAF) in EACs was > 10% and substantially larger than in ExAC[64] and in the European cohort from 1000 Genomes[27] using the following formula:

$$MAF_{EAC} > \sqrt{10 \times MAF_{reference}} \quad (3)$$

were $MAF_{reference}$ was the MAF in ExAC or 1000 Genomes.

Finally, variants were also removed if they exhibited an excess of heterozygosity determined as:

$$p_{Aa} > 1.04 - \sqrt{1.04 - 3.74 \times p_A \times p_a} \quad (4)$$

where $p_{Aa}$, $p_A$ and $p_a$ are the proportion of heterozygote variants, the major allele frequency and the minor allele frequency respectively. This formula was empirically chosen based on the principle of Hardy-Weinberg equilibrium[65]. The resulting variants were considered damaging if they were frameshift indels, introduced stopgains or stoplosses, or if defined damaging in at least five out of seven functional predictors as described above. Only rare damaging variants (MAF < 1% in ExAC[64] and in the Europeans from 1000 Genomes[27]) were further retained. A list of 152 cancer predisposition genes was obtained from the literature[26]. EAC patients that carried a rare damaging germline variant in one of these genes were considered to carry a cancer predisposition variant.

**Analysis of RNA sequencing data.** Purified total RNA was extracted from 92 EACs from the ICGC cohort and sequenced[16]. RNA sequencing reads were then aligned to human reference genome hg19 and expression values were calculated using Gencode v19. The summarise Overlaps function in the R GenomicAlignments package was used to count any fragments overlapping with exons (parameters mode = Union, singleEnd, invertStrand and inter.feature were set according to the library protocol, fragments = TRUE, ignore.strand = FALSE). Gene length was derived as the number of base pairs in the exons after concatenating the exons per gene in non-overlapping regions. FPKM (Fragments Per Kilobase Million) were calculated for each gene as:

$$FPKM = \frac{gene\ read\ count}{(library\ size/1000000) \times (gene\ length/1000)} \quad (5)$$

**Cell lines.** Overexpression and editing experiments were carried out using the FLO-1 esophageal adenocarcinoma cell line obtained from the ECACC General Cell Collection and CP-A (KR-42421) Barrett's Esophagus cells from ATCC (catalogue number CRL-4027). FLO-1 cells were grown at 37 °C and five per cent $CO_2$ in DMEM + 2 mM Glutamine + 10% FBS (Biosera) + 1/10,000 units of penicillin–streptomycin. CP-A cells were grown at 37 °C and five per cent $CO_2$ in Keratinocyte serum-free medium with 50 µg ml$^{-1}$ bovine pituitary extract and

5 ng ml$^{-1}$ recombinant human EGF (17005042, Thermo Fisher). For passaging of CP-A cells, 250 mg L$^{-1}$ soybean trypsin inhibitor in PBS was used (17075029, Thermo Fisher). Gene knockdown experiments were performed on OE19 cells obtained from the Francis Crick Institute cell services, OE33 cells obtained from the ECACC General Cell Collection and MFD1 cells obtained from the OCCAMS Consortium. OE19 and OE33 cells were grown in RPMI + 2 mM Glutamine + 10% FBS (Biosera) + 1/10,000 units of penicillin–streptomycin. MFD1 cells were grown in DMEM + 2 mM Glutamine + 10% FBS (Biosera) + 1/10,000 units of penicillin–streptomycin. All cells were maintained at 37 °C and five per cent CO$_2$, validated by short tandem repeat analysis and routinely checked for mycoplasma contamination.

**Gene overexpression.** The vectors pCMVHA E2F1[66] (Item ID 24225, Addgene), pLX_TRC317 (TRCN0000481188, Sigma-Aldrich) and pcDNA3.1 + /C-(K)-DYK (Clone ID: OHu19407D, Genscript) were used to induce *E2F1*, *MCM7*, and *PAK1* overexpression, respectively. FLO-1 cells were transfected according to the manufacturer's protocol, while CP-A cells were nucleofected following the Neon™ kit protocol (Thermo Fisher), with 2 pulses of 1200 V for 20 ms. Overexpressing cells were selected with either G481/Geneticin (*E2F1*, *PAK1*) or Puromycin (*MCM7*). Empty vectors carrying G418 (pcDNA3.1 + /C-(K)-DYK, Genscript) or Puromycin (Item ID 85966, Addgene) resistance were used as controls. The RNA from transfected cells was used to assess gene overexpression via quantitative RT-PCR using predesigned SYBR green primers (Sigma-Aldrich; Supplementary Table 3) and Brilliant III Ultra-Fast SYBR Green QRT-PCR Master Mix (Agilent Technologies). The average expression level across triplicates (*e*) was relativised to the average expression level of β-2-microglobulin (*c*):

$$r = e - c \qquad (6)$$

where *r* is the relative gene expression. The fold change (*fc*) between the relative gene expression after overexpression and the relative gene expression in the control condition ($r_c$) was calculated as:

$$fc = 2^{(r_c - rKD)} \qquad (7)$$

Each sample was assessed in triplicate and each experiment was repeated in biological duplicate.

**Gene editing.** To induce *ABI2* and *NCOR2* gene knock-out (KO) in FLO-1 cells, the vector-free CRISPR-mediated editing approach was used as previously described[32]. Briefly, cells were co-transfected using lipofectamine CRISPR max (Life technologies) with a 69-mer tracrRNA (Sigma-Aldrich), three gene-specific crRNAs (Sigma-Aldrich, Supplementary Table 3) and GeneArt Platinum Cas9 nuclease (Life technologies). To avoid off-target editing, all crRNAs used were verified to map only the gene of interest with a perfect match and additional hits in the genome with at least three mismatches. Control cells were transfected with the same protocol but using three non-targeting crRNAs. In CP-A cells, vector-free CRISPR-mediated editing of *ABI2*, *NCOR2* or *TP53* was performed by introducing two gene-specific crRNAs (Synthego, Supplementary Table 3) and GeneArt Platinum Cas9 nuclease (Life technologies) into the cells by nucleofection following the Neon™ kit protocol (Thermo Fisher), with 2 pulses of 1200 V for 20 ms. In all cases, gene editing was confirmed with Illumina MiSeq sequencing. The regions surrounding the targeted sites were amplified from genomic DNA of edited cells with primers containing Illumina adapters (Supplementary Table 3) using Q5 Hot Start High-Fidelity 2× Master Mix (New England Biolabs). DNA barcodes were added with a PCR reaction before pooling the samples for sequencing on Illumina MiSeq with the 250 base-pair paired-end protocol. Sequencing reads were merged into single reads and aligned to the human reference genome hg19 using BBMerge and BBMap functions of BBTools[67], obtaining an average of 78,864 aligned reads per experiment. SNVs and small indels in the regions corresponding to each crRNA (Supplementary Table 3) were called using the CrispRVariants package in R[68] and the percentage of edited alleles was estimated as the percentage of variant reads in each experiment.

**Gene knockdown.** Inducible gene knockdown was carried out using lentiviral pTRIPZ-TurboRFP (*MCM7*) or pSMART-TurboGFP (*PSMD3*) shRNA vectors (Dharmacon). For each gene, three shRNA vectors were tested (Supplementary Table 3). Virus was produced by co-transfecting HEK293T cells with pTRIPZ or pSMART constructs alongside psPAX2 and pMD2.G vectors (Addgene) using Fugene HD (Promega). Viral supernatant was collected at 24 and 48 h and used for two rounds of infection of OE19, OE33 or MFD1 cells, using 8 μg ml$^{-1}$ hexadimethrine bromide (Sigma-Aldrich). Infected cells were selected after 48 h with 2 μg ml$^{-1}$ puromycin for 7 days. To induce shRNA expression, cells were treated with 1 μg ml$^{-1}$ doxycycline (Sigma-Aldrich) for 16 h. Gene expression with or without doxycycline was assessed by qRT-PCR using predesigned SYBR green primers (Sigma-Aldrich; Supplementary Table 3). Cells with the highest level of knockdown were then sorted by FACS to isolate medium expressing cells (the middle 30% of cells based on TurboRFP or TurboGFP fluorescence). Gene expression after sorting was measured by qRT-PCR 24 h post-induction with 0–1 μg ml$^{-1}$ doxycycline, to determine the concentration of doxycycline required to reduce expression to levels equivalent to FLO-1 cells. The determined

concentrations of doxycycline used for proliferation assays were 0.05 μg ml$^{-1}$ for OE19 PSMD3 shRNA3, 0.25 μg ml$^{-1}$ for OE33 PSMD3 shRNA3, 0.25 μg ml$^{-1}$ for MFD1 PSMD3 shRNA3, 0.75 μg ml$^{-1}$ for MFD1 MCM7 shRNA3.

**Cell proliferation.** Cell proliferation was measured every 24 h for three or four days, starting three hours after seeding the cells (time zero) using crystal violet staining, CellTiter 96 Non-Radioactive Cell Proliferation Assay (Promega) or CellTiter-Glo Luminescent Cell Viability Assay (Promega). Briefly, $4.5 \times 10^3$ cells per well were seeded on 96-well plates in a final volume of 100 μl per well. For inducible shRNA-expressing cells, doxycycline was added 48 h prior to the start of the assay, and culture media replaced every 24–48 h with fresh media containing doxycycline. For the CellTiter 96 Non-Radioactive Cell Proliferation Assay, 15 μl of the dye solution was added into each well and cells were incubated at 37 °C for two hours. The converted dye was released from the cells using 100 μl of the solubilisation/Stop solution and absorbance was measured at 570 nm after one hour using the Paradigm detection platform (Beckman Coulter). For the CellTiter-Glo Luminescent Cell Viability Assay, 100 μl of the CellTiter-Glo reagent was added into each well and luminescence was measured after 30 min using the Paradigm detection platform (Beckman Coulter). For all proliferation assays, four replicates per condition were measured at each time point and each measure was normalised to the average time zero measure for each condition. Each experiment was repeated at least twice. Conditions were compared using the two-tailed Student's *t*-test.

**Flow cytometry.** EdU incorporation and MCM loading were assessed using a modified version of the protocol described in Galanos et al.[69]. Briefly, in each condition, $3 \times 10^6$ cells were pulsed for 30 min with 10 μM EdU (Invitrogen) before washing in 1% BSA/PBS. Chromatin fractionation was performed by incubating on ice for 10 min in CSK buffer (10 mM HEPES, 100 mM NaCl, 3 mM MgCl$_2$, 1 mM EGTA, 300 mM sucrose, 1% BSA, 0.2% Triton-X100, 1 mM DTT, cOmplete EDTA-free protease inhibitor cocktail tablets, Roche). Cells were then fixed in 4% formaldehyde/PBS for 10 min at room temperature before washing in 1% BSA/PBS. Cells were permeabilised and barcoded[70] by incubating in 70% ethanol containing 0–15 μg ml$^{-1}$ Alexa Fluor 488 (Thermo Fisher) for 15 min, then washed twice in 1% BSA/PBS. Barcoded cells were subsequently pooled before incubating in primary antibody (mouse monoclonal anti-MCM7: Santa Cruz Biotechnology sc-56324, or rabbit polyclonal anti-MCM3: Bethyl Laboratories A300–192A) diluted 1:100 in 1% BSA/PBS for 1 h. After washing in 1% BSA/PBS, samples were incubated for 30 min in secondary antibody (Alexa Fluor 555-conjugated donkey anti-mouse: A-31570, or donkey anti-rabbit: A-31572) diluted 1:500 in 1% BSA/PBS, then washed again in 1% BSA/PBS. EdU labelling with Alexa Fluor 647 azide was performed using Click-iT EdU flow cytometry assay kit (Invitrogen, C10424) following the manufacturer's instructions before washing samples in 1% BSA/PBS. Samples were then incubated in 1% BSA/PBS containing RNase and 10 mg ml$^{-1}$ DAPI for 15 min before analysing with a BD LSR II Fortessa flow cytometer (BD Biosciences). Lasers and filters used include: 407 nm laser with 450/50 bandpass filter; 488 nm laser with 505 longpass and 530/30 bandpass filters; 561 nm laser with 570 longpass and 590/30 bandpass filters; 640 nm laser with 670/14 bandpass filter. Compensation was performed manually with single colour controls, using BD FACSDiva software (BD Biosciences). FlowJo 10.3 software was used to analyse MCM loading onto chromatin and EdU incorporation. Cells were gated to remove debris using FSC-A/SSC-A, then gated to isolate singlets using DAPI-H/DAPI-A (Supplementary Fig. 9). The cells were then separated by gating the barcoded populations using 488-A/DAPI-A. Cells were finally separated into cell cycle gates (G1, S1-4, G2) based on EdU-647-A and DAPI-A (Supplementary Fig. 9), and the geometric mean fluorescence intensity was obtained for each channel (MCM-555 or EdU-647).

**ETHICS.** All subjects gave informed consent, and this study was registered (UKCRNID 8880) and approved by the Institutional Ethics Committees, Cambridgeshire 4 Research Ethics Committee (REC 07/H0305/52 and 10/H0305/1).

**Reporting Summary.** Further information on research design is available in the Nature Research Reporting Summary linked to this article.

## Data availability
WGS and RNA sequencing data can be accessed at the European Genome-phenome Archive using the accession numbers EGAD00001004775 and EGAD00001004776, respectively. TCGA data can be accessed through dbGaP accession number phs000178.v10.p8. Source data for Fig. 4–6 are provided as a Source Data file.

## Code availability
sysSVM is distributed as an R package under R 3.4.0 at https://github.com/ciccalab/sysSVM.

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

## Acknowledgements

We thank Stephanie Hills and John Diffley (The Francis Crick Institute) for their help with the experiments on E2F1 and MCM7 and for the discussion of results. We thank Valdone Maciulyte (The Francis Crick Institute) for help with the MiSeq library pre-paration. We are grateful for the support of the High Throughput Screening Facility, Flow Cytometry Facility, Advanced Sequencing Facility and Cell Services of the Francis Crick Institute. We thank Patricia Galipeau for providing the segmented copy number data of the Barrett's esophagus samples. We thank Alejandro Velazquez for the help with figures. The results published here are in whole or part based upon data generated by The Cancer Genome Atlas managed by the NCI and NHGRI. This work was supported by Cancer Research UK [C43634/A25487] and by the Cancer Research UK King's Health Partners Centre at King's College London. Computational analyses were done using the High-Performance Computer at the National Institute for Health Research (NIHR) Biomedical Research Centre based at Guy's and St Thomas' NHS Foundation Trust and King's College London. CY is supported by the Medical Research Council [MR/L001411/1]. Funding for sample sequencing was through the Oesophageal Cancer Clinical and Molecular Stratification (OCCAMS) Consortium as part of the International Cancer Genome Consortium and was funded by a programme grant from Cancer Research UK (RG66287). We thank the Human Research Tissue Bank, which is supported by the National Institute for Health Research (NIHR) Cambridge Biomedical Research Centre, from Addenbrooke's Hospital. Additional infrastructure support was provided from the CRUK funded Experimental Cancer Medicine Centre.

## Author contributions

F.D.C. conceived and directed the study. T.M. developed all the bioinformatics methods with the help of D.T. and J.N. and under the supervision of FDC, M.C. and C.Y. L.B. and E.F. performed the experiments with the help of MH and PS. TM, LB, EF, DT, JN, FDC analysed the data. JP helped with the analysis of RNA-Seq data. F.D.C., T.M., L.B. and E.F. wrote the manuscript. R.C.F., P.S., M.H., J.L. edited the manuscript. All authors approved the manuscript.

## Additional information

## The Oesophageal Cancer Clinical and Molecular Stratification (OCCAMS) Consortium

Ayesha Noorani[3], Paul A.W. Edwards[3,9], Rachael Fels Elliott[3], Nicola Grehan[3], Barbara Nutzinger[3], Caitriona Hughes[3], Elwira Fidziukiewicz[3], Jan Bornschein[3], Shona MacRae[3], Jason Crawte[3], Alex Northrop[3], Gianmarco Contino[3], Xiaodun Li[3], Rachel de la Rue[3], Annalise Katz-Summercorn[3], Sujath Abbas[3], Daniel Loureda[3], Maria O'Donovan[3,10], Ahmad Miremadi[3,10], Shalini Malhotra[3,10], Monika Tripathi[3,10], Simon Tavaré[9], Andy G. Lynch[9], Matthew Eldridge[9], Maria Secrier[9], Lawrence Bower[9], Ginny Devonshire[9], Sriganesh Jammula[9], Jim Davies[11], Charles Crichton[11], Nick Carroll[12], Peter Safranek[12], Andrew Hindmarsh[12], Vijayendran Sujendran[12], Stephen J. Hayes[13,14], Yeng Ang[13,15,16], Andrew Sharrocks[16], Shaun R. Preston[17], Sarah Oakes[17], Izhar Bagwan[17], Vicki Save[18], Richard J.E. Skipworth[18], Ted R. Hupp[19], J. Robert O'Neill[18,19], Olga Tucker[20,21], Andrew Beggs[20,22], Philippe Taniere[20], Sonia Puig[20], Timothy J. Underwood[23,24], Robert C. Walker[23,24], Ben L. Grace[23], Hugh Barr[25], Neil Shepherd[25], Oliver Old[25], James Gossage[26,27], Andrew Davies[26,27], Fuju Chang[26,27], Janine Zylstra[26,27], Ula Mahadeva[26], Vicky Goh[27], Grant Sanders[28], Richard Berrisford[28], Catherine Harden[28], Mike Lewis[29], Ed Cheong[29], Bhaskar Kumar[29], Simon L. Parsons[30], Irshad Soomro[30], Philip Kaye[30], John Saunders[30], Laurence Lovat[31], Rehan Haidry[31], Laszlo Igali[32], Michael Scott[33], Sharmila Sothi[34], Sari Suortamo[34], Suzy Lishman[35], George B. Hanna[36], Christopher J. Peters[36], Krishna Moorthy[36], Anna Grabowska[37], Richard Turkington[38], Damian McManus[38], David Khoo[39] & Will Fickling[39]

[9]Cancer Research UK Cambridge Institute, University of Cambridge, Cambridge CB2 0RE, UK. [10]Department of Histopathology, Addenbrooke's Hospital, Cambridge CB2 0QQ, UK. [11]Department of Computer Science, University of Oxford, Oxford OX1 3QD, UK. [12]Cambridge University Hospitals NHS Foundation Trust, Cambridge CB2 0QQ, UK. [13]Salford Royal NHS Foundation Trust, Salford M6 8HD, UK. [14]Faculty of Medical and Human Sciences, University of Manchester, Manchester M13 9PL, UK. [15]Wigan and Leigh NHS Foundation Trust, Wigan, Manchester WN1 2NN, UK. [16]GI science centre, University of Manchester, Manchester M13 9PL, UK. [17]Royal Surrey County Hospital NHS Foundation Trust, Guildford GU2 7XX, UK. [18]Edinburgh Royal Infirmary, Edinburgh EH16 4SA, UK. [19]Edinburgh University, Edinburgh EH8 9YL, UK. [20]University Hospitals Birmingham NHS Foundation Trust, Birmingham B15 2GW, UK. [21]Heart of England NHS Foundation Trust, Birmingham B9 5SS, UK. [22]Institute of Cancer and Genomic sciences, University of Birmingham, Birmingham B15 2TT, UK. [23]University Hospital Southampton NHS Foundation Trust, Southampton SO16 6YD, UK. [24]Cancer Sciences Division, University of Southampton, Southampton SO17 1BJ, UK. [25]Gloucester Royal Hospital, Gloucester GL1 3NN, UK. [26]Guy's and St Thomas's NHS Foundation Trust, London SE1 7EH, UK. [27]King's College London, London WC2R 2LS, UK. [28]Plymouth Hospitals NHS Trust, Plymouth PL6 8DH, UK. [29]Norfolk and Norwich University Hospital NHS Foundation Trust, Norwich NR4 7UY, UK. [30]Nottingham University Hospitals NHS Trust, Nottingham NG7 2UH, UK. [31]University College London, London WC1E 6BT, UK. [32]Norfolk and Waveney Cellular Pathology Network, Norwich NR4 7UY, UK. [33]Wythenshawe Hospital, Manchester M23 9LT, UK. [34]University Hospitals Coventry and Warwickshire NHS, Trust, Coventry CV2 2DX, UK. [35]Peterborough Hospitals NHS Trust, Peterborough City Hospital, Peterborough PE3 9GZ, UK. [36]Department of Surgery and Cancer, Imperial College, London W2 1NY, UK. [37]Queen's Medical Centre, University of Nottingham, Nottingham NG7 2UH, UK. [38]Centre for Cancer Research and Cell Biology, Queen's University Belfast, Belfast, Northern Ireland BT7 1NN, UK. [39]Queen's Hospital, Romford RM7 0AG, UK

