## [Peer Review File · Nature Communications]

Reviewers' comments:

Reviewer #1 (Remarks to the Author):

- The overall content is novel and well-structured. The potential of applying machine-learning to identify areas of opportunity by focusing on indications of perturbation rather than recurrence, which is highly dependent on sample-size, expands our ability to recognize patient-specific gene dependencies.
- The structure of the experiment and methods are represented clearly and appear to be reproducible.
- The validation of the effect of helper alterations (Section "Helper alterations contribute to cancer-related phenotypes and lead to dependence") should provide context on the comprehensive set of genes highlighted by SYSsvm. The given rationales for the helpers selected is thoughtful, but it is difficult to understand how representative the sampling is of the comprehensive set.
- The robustness of this analysis and the findings would be supported by additional experimentation outside of the current sample, which is limited to testing in cancer cells. An additional step could be taken to validate the perturbations associated with the genes highlighted by SYSsvm and whether they appear in normal cells. If successful, this would indicate clearly the role of these "helper" genes, as well as diminish the likelihood of a false positive.

Reviewer #2 (Remarks to the Author):

The manuscript entitled, "Patient-specific cancer genes contribute to recurrently perturbed pathways and establish novel vulnerabilities in esophageal adenocarcinoma" by Mourikis et al describe studies that used a machine learning algorithm to identify cancer genes in esophageal adenocarcinoma (EAC)s from 261 individual patients. They hypothesize that along with previously known driver events in EAC, there are additional complementary, or helper gene events. To assess the additional somatic alterations that may be present in individual patients they integrated mutations, copy number alterations and structural rearrangements and identify putative cancer genes that are patient-specific. The main finding is the EACs can be divided into six main clusters which may have implications for personalized therapy as many of the gene alteration converge on well-known cancer-related processes, including intracellular signalling, cell cycle regulation, DNA repair and Toll-like receptor signaling. They validated these findings in an independent cohort of 107 EAC and then for selected candidates experimentally examined their involvement in selected EAC biological processes such as cell proliferation.

The study is in general is interesting and the manuscript is well written. This work adds to our understanding of EAC and is potentially clinically relevant if as suggested targeting helpers turns out to influence cancer progression.

One concern is that "helper gene" status based on co-amplification suggested in this study and then "validated" in the TCGA and other EAC samples would most likely validate simply because amplification is very common in EAC and there are many large regions with many co-amplified genes. This made up 80% of the helper genes.

Reviewer #3 (Remarks to the Author):

The paper develops an interesting idea: the selection of cancer drivers based on consistency

across samples may be insufficient to detect other drivers that are rare and affect a few individuals (without entering in the conceptual problem on what is a driver).

Their proposal is to identify a large set of characteristics common to known drivers and use them to identify other genes with similar constellation of features as potential "rare" drivers.

The results indicate that the system is able to re-identify drivers, as well as many other "similar" genes.

To demonstrate the biological significance of the findings the authors apply a gene enrichment strategy that shows a general similarity between the two sets of genes and a clustering strategy that shows that the new genes contribute to the definition of the sub-classes of the selected cancer (esophageal adenocarcinoma).

First, it is not completely clear, how significant are these two experiments since by definition the genes have similar properties, including for example positions in the protein network, it can be expected that they will also have similar biological properties (enrichment) and will show coherence clustering properties.

The authors make an additional effort to test experimentally a hand full of genes, including some frequent and some very infrequently mutated ones. The results confirm that these genes have general oncogenic properties, but – as in all other similar cases – it is impossible to validate a statistical approach with a number of selected cases.

In summary, an interesting idea that is not sufficiently demonstrated in this paper.

Additionally, it will be interesting to know what is the genetic background of these cases, since germ line variants – presumably excluded from the analysis- could have a significant contribution to be combined with the proposed new oncogenes.

It will be also interesting to understand what are the properties that contribute to the classification. Indeed, it is surprising that this aspect has not been included in the paper, since this information will help to understand the origin of the predictions and can contribute to the definition of what is called a "helper" in the paper.

One other related aspect that also missed is the consideration of the concerted action of the mutations. With all the current developments on this area it could be expected a general consideration of the potential new mutations in the context of the other mutations including germ line variations.

Reviewer #1

1 - The validation of the effect of helper alterations (Section “Helper alterations contribute to cancer-related phenotypes and lead to dependence”) should provide context on the comprehensive set of genes highlighted by SYSsvm. The given rationales for the helpers selected is thoughtful, but it is difficult to understand how representative the sampling is of the comprehensive set.

RESPONSE: We agree with the reviewer that the experimental validation of few predictions, albeit confirmatory, may not be representative of the whole set of predictions. This is a common issue of doing experimental validation of selected predictions from new analytical methods (see also point 2 of Reviewer 3). To be as comprehensive as possible, we selected genes based on the following criteria:

- Span a broad range of alteration frequencies and validate frequently altered helpers (*E2F1* and *MCM7*) as well as rarely altered helpers (*NCOR2*, *ABI2*, and *PAK1*);
- Test the effect on EAC growth of loss of function alterations (*NCOR2*, *ABI2*) as well as gain of function/overexpression (*E2F1*, *MCM7*, *PAK1*);
- Validate key representatives of processes that define helper clusters (*E2F1* for cluster 2H and *MCM7* for cluster 4H) as well as of functional categories that are altered across all clusters (such as Rho GTPase effectors, represented by *ABI2* and *PAK1*)

As requested by the reviewer, we have now added further context and better explained how the selection was made (p. 11).

2 - The robustness of this analysis and the findings would be supported by additional experimentation outside of the current sample, which is limited to testing in cancer cells. An additional step could be taken to validate the perturbations associated with the genes highlighted by SYSsvm and whether they appear in normal cells. If successful, this would indicate clearly the role of these “helper” genes, as well as diminish the likelihood of a false positive.

RESPONSE: To address the reviewer’s comment, we performed two types of analyses:

1- We annotated the alterations of predicted helper genes in a recently published whole exome sequencing dataset of 82 samples of Barrett’s esophagus¹, which is a pre-cancerous condition associated with EAC. Interestingly, we found a significant enrichment in the proportion of damaged helper genes when compared to the rest of damaged genes (Figure 7B), suggesting that helper genes may be involved in the early phases of EAC development.

2- Since BE has a prevalence of damaging point mutations (Figure 7C), we modified *NCOR2* and *ABI2* in CP-A cells, which are derived from Barrett’s esophagus. The editing of *NCOR2* and *ABI2* led to significant increase of cell proliferation to levels comparable to those observed after editing TP53, the most frequently altered driver in EAC (Figure 7E).

The results of this analysis are described in the main text (p. 14) and in Figure 7.

Reviewer #2

1- One concern is that “helper gene” status based on co-amplification suggested in this study and then “validated” in the TCGA and other EAC samples would most likely validate simply because amplification is very common in EAC and there are many large regions with many co-amplified genes. This made up 80% of the helper genes.

RESPONSE: As the reviewer acknowledges, gene amplification is indeed a frequent event in EAC. However, we would like to further clarify that:

- 1- We did not use co-amplifications for the prediction of helpers. Gene gain is in fact only one of the 34 sysSVM features (Supplementary Table 1) and gene co-amplification is not explicitly used as a feature in the classifier;
- 2- We did not use the two independent cohorts (TCGA and Nones et al., 2014) to validate helper genes but to test whether sysSVM is able to assign higher scores to known drivers when they are not used as a training set. We showed that this is indeed the case (Supplementary Figure 3). Also for this analysis, co-amplification was not used as an explicit feature of the classifier and gene gain was one of the 34 sysSVM features;

To further check that, despite the massive gene amplification occurring in EAC, only a minor fraction of all amplified genes is predicted as helper genes, we performed two additional analyses:

- 1- We estimated the fractions of all amplified genes in EAC and of amplified genes that are predicted as helpers. We show that only a minority of all amplified genes is predicted as helper genes (Supplementary Figure 4);
- 2- We compared the number of EACs where the 952 helpers are amplified to the number of EACs where they are predicted as helpers. We show that this occurs only in 24.8% of cases (Supplementary Figure 4)

These two analyses confirm that gene amplification alone is not sufficient for a gene to be predicted as a helper gene and that our predictions are highly sample specific (*i.e.* the same amplified gene can be predicted as a helper in one sample but not in another, depending on how its score compares to the other altered genes in the two samples). Finally, we show that the contribution of copy number status to the final prediction models is comparable to that of other features (Figure S3 and point 4 of Reviewer 3).

We summarise these further analyses in the manuscript (p. 7) and in Supplementary Figure 4.

Reviewer #3

1- To demonstrate the biological significance of the findings the authors apply a gene enrichment strategy that shows a general similarity between the two sets of genes and a clustering strategy that shows that the new genes contribute to the definition of the sub-classes of the selected cancer (esophageal adenocarcinoma). First, it is not completely clear, how significant are these two experiments since by definition the genes have similar properties, including for example positions in the protein network, it can be expected that they will also have similar biological properties (enrichment) and will show coherence clustering properties.

RESPONSE: While the reviewer correctly points out that helpers and drivers have similar molecular and systems-level properties, this does not necessarily imply that their functional properties would be the same. In fact, the sysSVM classifier does not take into account gene function in its classification. For example, the positions of the proteins in the protein-protein interaction network mentioned by the reviewer are encoded in the classifier as node degree and betweenness and whether the protein is central and/or a hub of the network (Supplementary Table 1). However, the labels of its interactors are not part of the classifier. In other words, the classifier knows the network property of each protein but not with whom it interacts.

2- The authors make an additional effort to test experimentally a hand full of genes, including some frequent and some very infrequently mutated ones. The results confirm that these genes have general oncogenic properties, but – as in all other similar cases – it is impossible to validate a statistical approach with a number of selected cases.

RESPONSE: We agree with the reviewer that the experimental validation of few predictions, albeit confirming the predictions, may not be representative of the comprehensive set of predicted helpers (see also point 1 of Reviewer 1). As acknowledged by the same reviewer, this is a common issue of doing experimental validation on selected genes. On the other hand, the alternative of experimentally validating almost 1,000 genes is unfeasible.

We were as comprehensive as we could be in the criteria for gene selection and tried:

- To span a broad range of alteration frequencies and validate frequently altered helpers (*E2F1* and *MCM7*) as well as rarely altered helpers (*NCOR2*, *ABI2*, and *PAK1*);
- To validate the effect on EAC growth of loss of function alterations (*NCOR2*, *ABI2*) as well as gain of function/overexpression (*E2F1*, *MCM7*, *PAK1*);
- To validate key representatives of processes that define helper clusters (*E2F1* for cluster 2H and *MCM7* for cluster 4H) as well as of functional categories that are altered across all clusters (such as Rho GTPase effectors, represented by *ABI2* and *PAK1*)

We have now added further context and better explained how the selection was made (p. 11).

3- Additionally, it will be interesting to know what is the genetic background of these cases, since germ line variants – presumably excluded from the analysis- could have a significant contribution to be combined with the proposed new oncogenes.

RESPONSE: To address this point, we identified samples with rare damaging germline variants in 152 known cancer predisposition genes². As expected for EAC³, we did not observe enrichment in any of these genes compared to healthy controls (Supplementary Figure 8).

We then tested for associations between the damaged predisposition genes and the patient clusters. We found a significant depletion of predisposition variants in cluster 4H, which is associated with somatic alterations in DNA replication genes (Figure 3A).

We describe and comment on these results in the text (pp. 9-11). We also added new panels in Figure 3A and Supplementary Figure 8.

4- It will be also interesting to understand what are the properties that contribute to the classification. Indeed, it is surprising that this aspect has not been included in the paper, since this information will help to understand the origin of the predictions and can contribute to the definition of what is called a “helper” in the paper.

RESPONSE: To address this point, we implemented the Recursive Feature Elimination (RFE) method, as described in⁴. RFE first defines the best set of parameters for each kernel. Secondly, it trains the classifier in each kernel and computes the weight (w) of each feature. Thirdly, it recursively removes the feature with the smallest value of w^2 until no feature remains. This analysis showed a few interesting features of sysSVM:

- 1- The top-ranking features differed between linear (linear and polynomial) and non-linear (radial and sigmoid) kernels. In particular, categorical features were the top contributors for the linear kernels, while non-linear kernels had a mixed contribution of categorical and continuous features (Supplementary Figure 3). This is likely to reflect differences across kernels and supports the integration of multiple kernels in sysSVM as a mean to capture different regions of the feature space.
- 2- No feature had zero weight in all kernels, indicating that all of them contribute to the final gene classification.
- 3- Despite the high prevalence of copy number gain in EAC, gene amplification was not the highest-ranking feature in any of the kernels (see also point 1 of Reviewer 2).

We describe these results in the main text (p. 6), in the methods (p. 21) and in Supplementary Figure 3.

5- One other related aspect that also missed is the consideration of the concerted action of the mutations. With all the current developments on this

area it could be expected a general consideration of the potential new mutations in the context of the other mutations including germ line variations.

RESPONSE: We agree that taking into account the concerted action of mutations is important to fully understand the landscape of cancer driver events. We sought to investigate this in several ways:

- We examined the co-occurrence of known drivers and helpers and we describe these results in the main text (pp. 9-10) and in Figure 2D.
- Due to the fact that the majority of helpers are altered in only one or very few samples (Figure 1F), it is not possible to assess the association between helpers. However, we investigated the convergence of different helper genes to perturbations of similar biological pathways. This led to the identification of the 6 helper clusters (Figure 2C and Supplementary Figure 6).
- We measured the association between germline variations in 152 known cancer predisposing genes and helper clusters. We found a significant depletion of predisposition variants in cluster 4H (Figure 3A). We describe and comment on these results in the text (p. 10-11), Figure 3A and Supplementary Figure 8.

References

1. Galipeau PC, *et al.* NSAID use and somatic exomic mutations in Barrett's esophagus. *Genome Med* **10**, 17 (2018).
2. Huang KL MR, Wu Y, Ritter DI, Wang J, Oh C, Paczkowska M, Reynolds S, Wyczalkowski MA, Oak N, Scott AD, Krassowski M, Cherniack AD, Houlahan KE, Jayasinghe R, Wang LB, Zhou DC, Liu D, Cao S, Kim YW, Koire A, McMichael JF, Huchtagowder V, Kim TB, Hahn A, Wang C, McLellan MD, Al-Mulla F, Johnson KJ; Cancer Genome Atlas Research Network, Lichtarge O, Boutros PC, Raphael B, Lazar AJ, Zhang W, Wendl MC, Govindan R, Jain S, Wheeler D, Kulkarni S, Dipersio JF, Reimand J, Meric-Bernstam F, Chen K, Shmulevich I, Plon SE, Chen F, Ding L. Pathogenic Germline Variants in 10,389 Adult Cancers. *Cell* **173**, (2018).
3. Anna M.J. van Nistelrooij WNMD, Anja Wagner, Manon C.W. Spaander, J. Jan B. van Lanschot, Bas P.L. Wijnhoven. Hereditary Factors in Esophageal Adenocarcinoma. *Gastrointestinal Tumors* **1**, (2014).
4. Guyon I, Weston J, Barnhill S, Vapnik V. Gene Selection for Cancer Classification using Support Vector Machines. *Machine learning* **46**, 389-422 (2002).

REVIEWERS' COMMENTS:

Reviewer #1 (Remarks to the Author):

The work done by Prof. Ciccarelli et al. is significant in two aspects: 1) their findings with regards to the relative impact and prevalence of previously less understood genes causing perturbation, and 2) their methodology in applying a machine learning algorithm to the identification of helper genes in EAC. They also used the recurrence of process perturbation to stratify the 261 EACs into six clusters that show distinct molecular and clinical features and suggest differential response to targeted treatment.

The team's work in designing a study centered on machine learning is of interest across fields. Specifically, this study is an excellent example of how data science can be applied to research topics that have been left unpursued due to problems of statistical significance. The framework and tools applied can be similarly leveraged against other constraints of frequency, sample size, etc. The study also speaks to the future of research, enabled by well-maintained centralized databases such as the International Cancer Genome Consortium. The application of data science would be impossible without the extensive body of research that has been shared.

The study is replicable. the algorithm is shareable. The statistical principles applied are sound.

Reviewer #2 (Remarks to the Author):

The authors have adequately addressed my concerns.

Reviewer #3 (Remarks to the Author):

The revised version has addressed to a sufficient level my concerns.
I have only two remaining points.

"While the reviewer correctly points out that helpers and drivers have similar molecular and systems-level properties, this does not necessarily imply that their functional properties would be the same. In fact, the sysSVM classifier does not take into account gene function in its classification. For example, the positions of the proteins in the protein-protein interaction network mentioned by the reviewer are encoded in the classifier as node degree and betweenness and whether the protein is central and/or a hub of the network (Supplementary Table 1). However, the labels of its interactors are not part of the classifier. In other words, the classifier knows the network property of each protein but not with whom it interacts."

This is not really addressing the point I wanted to make.

The point was that nodes of the network with similar topological properties (not labels) will tend to have some basic similarity (some functions tend to be connected in specific ways) or at least a correlation would be expected between network and biological properties. Is this the case? Do the nodes with similar network properties have similar biological properties?

Is the similarity level a confusing factor for the proposed analyses?

Fig 7b is important. It should be better explained in the text and the (small but apparently significant!) differences more clearly explained.

Reviewer #3 (Remarks to the Author):

The revised version has addressed to a sufficient level my concerns. I have only two remaining points.

"While the reviewer correctly points out that helpers and drivers have similar molecular and systems-level properties, this does not necessarily imply that their functional properties would be the same. In fact, the sysSVM classifier does not take into account gene function in its classification. For example, the positions of the proteins in the protein-protein interaction network mentioned by the reviewer are encoded in the classifier as node degree and betweenness and whether the protein is central and/or a hub of the network (Supplementary Table 1). However, the labels of its interactors are not part of the classifier. In other words, the classifier knows the network property of each protein but not with whom it interacts."

THIS is not really addressing the point I wanted to make.

The point was that nodes of the network with similar topological properties (not labels) will tend to have some basic similarity (some functions tend to be connected in specific ways) or at least a correlation would be expected between network and biological properties. Is this the case? Do the nodes with similar network properties have similar biological properties?

Is the similarity level a confusing factor for the proposed analyses?

RESPONSE: To address the reviewer comment, we have performed a pathway enrichment analysis of the central hubs of the network (top 25% most connected and central nodes, $n = 2,608$ proteins). This resulted in 528 enriched pathways (FDR = 0.01) out of 1,155 total Reactome pathways (46%), indicating that there is large diversity in the functions of central hubs. Therefore, nodes with similar topological properties do not necessarily have similar biological properties.

To further check whether the functions of known cancer genes could in any way bias the functional properties of central hubs, we performed a pathway enrichment analysis of the central hubs encoded by known cancer genes ($n = 252$ proteins). This resulted in 158 enriched pathways (FDR = 0.01). Of these, 135 were shared with all central hubs (25%), indicating that cancer-related functions constitute only a small fraction of pathways enriched in central hubs.

From both these observations, we conclude that the enrichment of newly discovered helpers in cancer-related pathways is not due to a bias in the protein-protein interaction network, but it likely reflects their genuine similarity to known cancer genes.

We describe these further analyses at p.6.

Fig 7b is important. It should be better explained in the text and the (small but apparently significant!) differences more clearly explained.

RESPONSE: We further discuss the importance of this observation at p.14 (Results) and p.16 (Discussion).